# Towards Understanding the Feasibility of Machine Unlearning

## Abstract

In light of recent privacy regulations, machine unlearning has attracted significant attention in the research community. However, current studies predominantly assess the overall success of unlearning approaches, overlooking the varying difficulty of unlearning individual training samples. As a result, the broader feasibility of machine unlearning remains under-explored. This paper presents a set of novel metrics for quantifying the difficulty of unlearning by jointly considering the properties of target model and data distribution. Specifically, we propose several heuristics to assess the conditions necessary for a successful unlearning operation, examine the variations in unlearning difficulty across different training samples, and present a ranking mechanism to identify the most challenging samples to unlearn. We highlight the effectiveness of the Kernelized Stein Discrepancy (KSD), a parameterized kernel function tailored to each model and dataset, as a heuristic for evaluating unlearning difficulty. Our approach is validated through multiple classification tasks and established machine unlearning algorithms, demonstrating the practical feasibility of unlearning operations across diverse scenarios.

## 1 Introduction

Machine Unlearning (MU) (Cao & Yang, 2015) refers to a process that enables machine learning (ML) models to remove specific training data and revert corresponding data influence on the trained models while preserving the models' generalization. As many countries and territories have promulgated their Right to be Forgotten regulations [1], entitling individuals to revoke their authorization to use their data for machine learning (ML) model training, the demand of MU raised significant interest in the machine learning research community, leading to various types of unlearning approaches, often achieved by either data reorganization (Graves et al., 2021; Gupta et al., 2021; Tarun et al., 2023) or model manipulation (Guo et al., 2020; Warnecke et al., 2021).

Although existing machine unlearning studies vary based on diverse theoretical foundations, their performance evaluation metrics used are generally common, including 1) Data Erasure Completeness, 2) Unlearning Time Efficiency, 3) Resource Consumption, and 4) Privacy Preservation (Xu et al., 2024; Yang & Zhao, 2023; Shaik et al., 2023). While such quantitative evaluations may suffice for comparing MU methods from various perspectives, they often fall short in assessing the effectiveness of data removal requests for individual data points, resulting in a discrepancy between actual unlearning outcomes and expectations in real-world applications. In practice, some data points are inherently harder to unlearn than others, where such variability may stem from intrinsic factors, such as the augmented data distribution under a trained machine learning model, regardless of the specific unlearning algorithm applied.

While quantifying the difficulty of unlearning operations (Marchant et al., 2022; Pawelczyk et al., 2024) is underexplored, one may come up with a handy metric constructed on top of several obvious factors, such as model complexity, prediction confidence, and the nature of input data features. Indeed, the most straightforward criterion of success of unlearning is to measure the distance of the data sample from the model's decision boundaries (Liu et al., 2023b; Chen et al., 2023); data points closer to the decision boundary are more likely to be forget-able. Unfortunately, we show in this paper that although the factors above have a certain degree of positive correlation with the difficulty of unlearning, they are often accompanied by noise, leaving them lacking credibility.

---

[1]CCPA in California, GDPR in Europe, PIPEDA in Canada, LGPD in Brazil, and NDBS in Australia.

This paper paves the first step on understanding the feasibility of machine unlearning (regardless of specific unlearning approaches) by analyzing and quantifying the difficulty of unlearning the training data points on a trained model, such that identifying the criteria contributing to this process. Instead of falling back to predictive confidence-based approaches (Liu et al., 2023b; Chen et al., 2023), we propose to measure the difficulty of unlearning through model-augmented data distribution. Specifically, we show that trained ML models can uniquely define parameterized kernel functions $k_\theta(\cdot, \cdot)$ over training data points, which allow to express the distribution of training samples conditioned on trained model $f_\theta$ in the form of a correlation graph. Due to the generalization of machine learning models, training samples that have strong connections with other samples may face the challenge of unlearning. While there are many choices of parameterized kernel functions, we show that the kernel defined on Kernelized Stein Discrepancy (KSD) (Liu et al., 2016; Chwialkowski et al., 2016) shows a unique advantage in terms of accurately reflecting a trained model's characteristics given its training samples. By analyzing the property of KSD, we show that there is a set of reliable scoring metrics based on KSD that can efficiently measure the difficulty of unlearning. With the proposed evaluation metrics, one may reduce unnecessary machine unlearning operations when data points are determined to be infeasible to unlearn.

## 2 Preliminaries

### 2.1 Machine Unlearning Definition

Machine Unlearning (MU) is a process of removing specific subsets of training data (along with their influence) from a trained model (Cao & Yang, 2015; Bourtoule et al., 2021).

Formally, consider a training dataset $\mathcal{D}_t = \{(\mathbf{x}_i, y_i)\}$ ($t$ is refering to training data) comprising $n$ samples, where $\mathbf{x}_i$ and $y_i$ represent the $i^{th}$ data's features and corresponding label respectively. We define two subsets of the dataset for clarity as follows: Let $\mathcal{D}_f \subseteq \mathcal{D}_t$ denote the subset of data designated to be forgotten (a.k.a *forget set*), and $\mathcal{D}_r \subseteq \mathcal{D}_t$ denote the remaining data (a.k.a *remaining set*), such that $\mathcal{D}_f \cup \mathcal{D}_r = \mathcal{D}_t$ and $\mathcal{D}_f \cap \mathcal{D}_r = \emptyset$.

Given an target predictive model $f_\theta$ with parameters $\theta$, the common expectation of machine unlearning operation are of adjusting $\theta$ to a modified parameter set $\vartheta$ such that:

- Increasing of model's error on the forget set $\mathcal{L}_\vartheta(\mathcal{D}_f)$.
- Maintaining original model's error on the remaining set $\mathcal{D}_r$ such that

$$\|\mathcal{L}_\theta(\mathcal{D}_r) - \mathcal{L}_\vartheta(\mathcal{D}_r)\| < \epsilon, \tag{1}$$

where $\epsilon$ denotes a tolerable performance degradation threshold and $\mathcal{L}$ denotes the loss.

### 2.2 Research Track of Machine Unlearning

The simplest solution for unlearning is to retrain the model from scratch with the *remaining data* after removing the *forget data*. However, even with tricks such as partial retraining (Bourtoule et al., 2021), that works on decomposable, partial model components, retraining remained resource-intensive. To mitigate the computational overhead of retraining, unlearning operations are often approximated. In particular, Fine-Tuning based approaches (Warnecke et al., 2021; Golatkar et al., 2020) suggest to continue training the model on the remaining data such that forget data can be naturally flashed out. Alternatively, Gradient Ascent approach (Graves et al., 2021) (referred as NegGrad in (Kurmanji et al., 2024)) adjust the model's weights in the direction of the gradient to increase the model's error on the data intended for forgetting.

Recent studies have frequently utilized the Newton update as a fundamental step for removing data influence (Guo et al., 2020; Golatkar et al., 2020; Peste et al., 2021; Sekhari et al., 2021). These methods typically leverage the Fisher Information Matrix (FIM) to gauge the sensitivity of the model's output to perturbations in its parameters. For example, Fisher Forgetting (Golatkar et al., 2020) employs a scrubbing approach where noise is added to parameters based on their relative importance in distinguishing the forget set from the remaining data set. Mehta et al. (Mehta et al., 2022) employs conditional independence coefficient to identify sufficient sets of parameters for targeted unlearning.

As machine unlearning often involve privacy protection regulations, some methods also incorporated the principles of differential privacy (DP) (Abadi et al., 2016) to ensure the unlearning outcome does not inadvertently reveal information about the data that has been removed. Izzo (Izzo et al., 2021; Zhang et al., 2024) adhere to the DP framework to ensure a high probabilistic similarity between models before and after unlearning. Golatkar et al. (2020) introduced the concept of certified unlearning, grounded in information theory and specifically tailored to the Fisher Information Matrix. Certified Minimax Unlearning (Liu et al., 2023a) has developed an algorithm specifically for minimax models. This method removes data influences through a total Hessian update and incorporates the Gaussian Mechanism to achieve $(\epsilon, \delta)$-minimax unlearning certification, ensuring a balance between data removal and model integrity. (Chourasia & Shah, 2023) propose a data deletion technique that ensures the privacy of deleted records. (Ullah et al., 2021) investigated the machine unlearning in the context of SGD and streaming removal requests, but ensured that their method is differentially private. DP algorithms provide the upper bound for the unlearning scheme, but they don't guarantee the full unlearning of requested data(Nguyen et al., 2022).

Other research efforts extend into various domains of unlearning, such as knowledge distillation (Chundawat et al., 2023), selective forgetting for lifelong learning (Shibata et al., 2021), federated unlearning (Che et al., 2023), online unlearning (Li et al., 2021; Chen et al., 2019), and exploring adversarial attacks using machine unlearning methods (Wei et al., 2023; Di et al., 2022). Instead of aiming to follow privacy protection regulation, those approaches focus on reducing the vulnerability of models to adversarial attacks by forgetting training examples (Jagielski et al., 2023).

The majority of the literature on machine unlearning primarily concentrates on the development of unlearning algorithms or unlearning approximation techniques for selectively forgetting data from a trained model. As such, the corresponding evaluation metrics are designed to favor the performance difference between algorithms on highly aggregated level (e.g. success rate). In fact, a common assumption underlying much of this research is that unlearning operations are universally feasible for all data points within a dataset, where effectiveness of unlearning will behave consistently across different datasets. This assumption often overlooks the potential variability in unlearning efficacy due to differences in data characteristics or model dependencies, suggesting a need for more nuanced studies that evaluate the specific conditions under which MU can be effectively implemented. Recently, one line work (Thudi et al., 2022) has questioned the performance of unlearning approximation algorithms w.r.t the exact unlearning and whether or not these methods can successfully prove the absence of certain data points during training. From this recent research, we can observe that absence of a comprehensive study that comprehensively investigates the feasibility of unlearning was present.

## 3 FACTORS THAT AFFECT DIFFICULTY OF UNLEARNING

In this section, we summarize the main factors that impact effectiveness of machine unlearning with corresponding data analysis and intuition justifications.

Intuitively, to quantify success of unlearning operation, one may evaluate the quality of the unlearned model $\vartheta$ with respect to the common expectation mentioned in previous section. Particularly, when the forget set contains a single sample $(\mathbf{x}_f, y_f)$, we expect the prediction of the unlearned model on that data point to be same as of a model trained without the data point (a.k.a Certified Unlearning (Guo et al., 2020)), reflecting the model's ignorance to forgotten example. However, it is often impractical to verify the success of unlearning by comparing to a re-trained model; such comparison lose the practical value of unlearning. Hence, evaluation criteria of successful unlearning operation in practice often falls back to measuring 1) the shift of prediction (e.g. altering predicted classification label) along with 2) the model performance preservation criterion (See Eq. 1). Unfortunately, the two criteria mentioned above are often a pair of trade-off, resulting the following factors that jointly pose challenges for defining the difficulty of unlearning.

1. **Size of Unlearning Expansion:** Altering prediction outcome of target sample may negatively impact model prediction on similar samples. When a guaranteed unlearning is desired, one might need to expand unlearning operation to a broader training sample set (the similar data samples) such that unlearning of target sample with respect to decision shift can be successful (Chen et al., 2023). While selection of unlearning algorithm may vary the size of the expansion, the need of expanding forget set for guaranteed unlearning remains consistent.

2. **Resistance to Membership Inference Attack (MIA):** In MIA (Chen et al., 2021; Golatkar et al., 2021; Song et al., 2019), adversaries use the model's outputs, such as confidence scores, to ascertain if a specific data point was part of the training set, without needing direct access to the model's internal parameters. With identical unlearning operations (e.g. number of unlearning step, learning rate, and other hyper-parameters), data points often show different level of resistance to the MIA, which is an alternative factor of measuring the difficulty of unlearning.

3. **Geometric Distance to Decision Boundary:** The distance of a data point to the decision boundary of a model often closely associated with the predictive confidence. A data point with lower predictive confidence (larger uncertainty) is closer to the decision boundary (Nguyen & Smeulders, 2004; Li et al., 2023). A data with higher uncertainty might be easier to unlearn compared to those with high predictive confidence (Chen et al., 2023). However, predictive confidence does not account for the relationships among similar examples. A data point with low predictive confidence might be deeply embedded within a complex region of the decision boundary (Kienitz et al., 2022), requiring significant modifications to unlearn. Hence, it remains to be a noisy index.

4. **Tolerance of Performance Shift:** For an easily unlearn-able data point, guaranteed unlearning is achievable with minimal impact on the model's predictive performance (Liu et al., 2024a). Here the model performance measurement is on remaining, and test dataset rather than remaining training dataset (as given in Equation 1). Conversely, an data point that is hard to unlearn would require significant changes to the model parameters, leading to a substantial deviation in performance metrics.

5. **Number of Unlearning Steps:** Number of Unlearning Steps evaluates the computational efficiency of the unlearning operations, indicating how quickly the model can be updated to forget a specified data. For a given unlearning algorithm, the metric can be approximated through wall clock duration (Nguyen et al., 2022).

6. **Distance of Parameter Shift (DPS):** Instead of using model performance changes as heuristic, DPS directly probe the layer-wise and activation-wise parameter shift distances. Layer-wise distance measures the weight differences between the unlearned and original models, while activation-wise distance assesses their activation differences given the same input (Golatkar et al., 2020; Tarun et al., 2023).

With the identified *unlearning feasibility factors* that affect difficulty of unlearning, one may seek for a coherent scoring systems that can rank training data points from the most easy to the most challenging for unlearning operations, as we will discuss in the next section.

## 4    SCORING UNLEARNING DIFFICULTY

We summerize the unlearning difficulty factors into two major groups namely 1) data points with/without strong ties (factor 1, 4-6) and 2) predictive confidence (factor 2-3). Our aim is to develop a unlearning difficulty scoring metric that jointly considers these two classes of factors.

Before moving forward, we briefly describe the Kernelized Stein Discrepancy (KSD) (Liu et al., 2016; Chwialkowski et al., 2016) to facilitate the later discription.

### 4.1    KERNEL STEIN DISCREPANCY (KSD)

The KSD provides a robust measure to evaluate the goodness-of-fit between a machine learning model and its training dataset. KSD originates from a mathematical theorem known as Stein's Identity (Kattumannil, 2009), which posits that for a smooth distribution $p(x)$ and a function $\phi(x)$ satisfying the condition $\lim_{||x|| \to \infty} p(x)\phi(x) = 0$, the following equation holds for any function $\phi$:

$$\mathbb{E}_{x \sim p}\big[\phi(x)\nabla_x \log p(x) + \nabla_x \phi(x)\big] = \mathbb{E}_{x \sim p}\big[\mathcal{A}_p \phi(x)\big] = 0, \quad \forall \phi \tag{2}$$

where $\mathcal{A}_p$ denotes the Stein operator, encapsulating the distribution $p(x)$ in terms of derivatives.

Kernelized Stein Discrepancy, derived from Stein's Identity, quantifies the difference between two distributions by defining the search for an optimal function $\phi$ in $\mathcal{F}$ through an appropriate kernel function $\kappa$. Concretely, KSD is defined as:

$$\mathbb{S}(q, p) = \mathbb{E}_{x,x' \sim q}[\kappa_p(x, x')] \tag{3}$$

where $\kappa_p(x, x') = \mathcal{A}_p^x \mathcal{A}_p^{x'} k(x, x')$ and can operate with any arbitrary kernel function $k(x, x')$. This formulation allows for a more feasible application in various practical scenarios, making KSD a valuable tool for assessing model-data compatibility. Indeed, in the goodness-of-fitness context, the $p$ distribution often represent predictive distribution of model parameterized with $\theta$, while $q$ distribution denotes the data distribution (Yang et al., 2018; Jitkrittum et al., 2020).

By examining the parameterized kernel defined on pairs of training data points as follows

$$
\begin{aligned}
\kappa_\theta((\mathbf{x}_a, y_a), (\mathbf{x}_b, y_b)) &= \mathcal{A}_\theta^a \mathcal{A}_\theta^b k(a, b) \\
&= \nabla_a \nabla_b k(a, b) && \Rightarrow \text{Raw Feature Similarity} \\
&+ k(a, b) \nabla_a \log P_\theta(a) \nabla_b \log P_\theta(b) && \Rightarrow \text{Score Similarity} \\
&+ \nabla_a k(a, b) \nabla_b \log P_\theta(b) + \nabla_b k(a, b) \nabla_a \log P_\theta(a), && \Rightarrow \begin{cases} \text{Mutual Influence of} \\ \text{Prediction Shifts} \end{cases}
\end{aligned} \tag{4}
$$

we note it can serve as base of evaluating the difficulty of machine unlearning well given it measures both model augmented data similarity (tie) and hypothetical predictive confidence shifts (as gradient/score).

### 4.2 Unlearning Difficulty Scoring with KSD

While the parameterized kernel defined by KSD has potential to distinguish data points with respect to the difficulty of unlearning, KSD itself is not sufficient to be a scoring function since it is defined on a pair of data points. To corporate Stein Kernel to rank datapoints' unlearning difficulty, we need to aggregate pair-wise kernel values for each data point such that

$$\text{Agg}(\{\kappa_\theta((\mathbf{x}_i, y_i), (\mathbf{x}_j, y_j)) | \forall j \text{ s.t. } (\mathbf{x}_j, y_j) \in D_t\}), \tag{5}$$

which requires examining the distribution of Stein Kernel values for targeted removal data point $i$.

We analyzed several variations of scoring metrics based on the Kernelized Stein Discrepancy (KSD) measure, as outlined below. The evaluation of each training data point yields a scalar value, which enables the ranking of unlearning difficulty across samples.

- **Marginalized KSD (MKSD):** The MKSD scoring comes from a simple idea of aggregation over the KSD kernel values between a data point and all others in the dataset in an computationally easy way

$$\text{Agg}_{\text{MKSD}}((\mathbf{x}_i, y_i)) = \Sigma_{j=1}^n \kappa_\theta((\mathbf{x}_i, y_i), (\mathbf{x}_j, y_j)) \tag{6}$$

A higher MKSD score suggests greater tie of a sample and a larger "resistance set", implying that more extensive sets of the training samples need to be unlearned alongside the target data point to guarantee the success of unlearning.

- **Marginalized Standardized KSD (MSKSD):** Intuitively, a data point surrounded by a large number of similar points poses more challenges in unlearning due to its strong connections within the dataset. To effectively highlight such data points, we define the MSKSD which aggregates these standardized exponential values. Similar to MKSD, a higher MSKSD value suggesting a higher challenge in unlearning these particular instances.

$$\text{Agg}_{\text{MSKSD}}((\mathbf{x}_i, y_i)) = \sum_{j=1}^n e^{\hat{\kappa}_\theta((\mathbf{x}_i, y_i), (\mathbf{x}_j, y_j))}, \tag{7}$$

where we denote standardized Stein values as $\hat{\kappa}$. The standardization operation, that re-assigning kernel values into standard Guassian $\mathcal{N}(0, 1)$, can reduce the influence of magnitude of kernel values such that the aggregation will be influenced by the skewness of smoothed values ($\hat{\kappa}$).

- **Stein Score Norm (SSN)**: Data points that are proximate to the decision boundary exhibit higher gradient magnitudes, making it easier to adjust the boundary in response to these points compared to those located deeper within each class. We propose that data points with high SSN (see Equation 8) are positioned away from the dense centers of their classes, making them prime

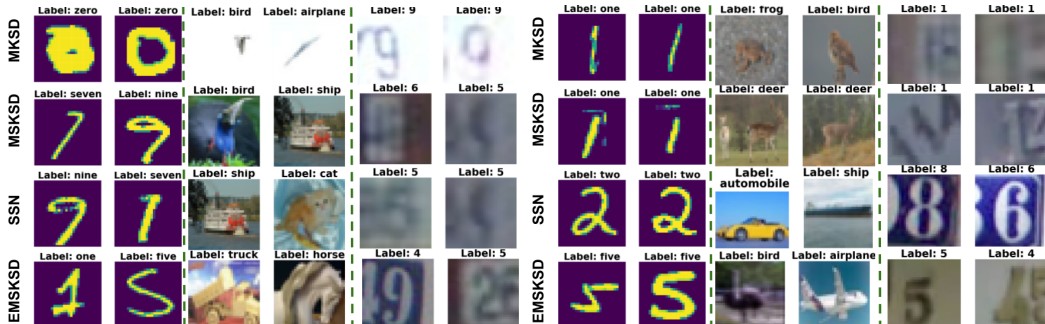

(a) Samples are easy to be unlearned        (b) Samples are hard to be unlearned

Figure 1: Top-2 most easy and hard training samples for machine unlearning flagged by the four proposed unlearning difficulty scoring functions. While the samples flagged varies based on the different scoring function, we note the easily unlearnable samples are often less representative to the class label, while hardly unlearnable samples are often typical.

candidates for unlearning. By identifying and ranking data points with the highest Stein Score Norms, we can effectively pinpoint those that are closer to the decision boundary.

$$\text{Agg}_{\text{SSN}}((\mathbf{x}_i, y_i)) = ||\nabla_\theta \log P_\theta(\mathbf{x}_i, y_i)|| \tag{8}$$

- **Entropy Marginalized Standardized KSD (EMSKSD)**: Uncertainty in the predictive confidence of a datapoint can be an indication of its' closeness to decision boundary. In EMSKSD, we incorporating datapoints predictive uncertainty with the similarity scoring:

$$\text{Agg}_{\text{EMSKSD}}((\mathbf{x}_i, y_i)) = \frac{\text{Agg}_{\text{MSKSD}}((\mathbf{x}_i, y_i))}{H(\hat{\mathbf{y}}_i)}$$

$$H(\hat{\mathbf{y}}_i) = -\sum_{l=1}^{n} \hat{\mathbf{y}}_{i,l} \log(\hat{\mathbf{y}}_{i,l}) \quad \hat{\mathbf{y}}_{i,l} = f_\theta(\mathbf{x}_i) \tag{9}$$

Here, $H(\hat{\mathbf{y}}_i)$ is the entropy of the predicted probabilities. EMSKSD can obtain points close to the decision boundary with small number of correlated samples.

The scoring metrics above are not comparable based on our analysis as shown in Figure 1; they flag different types of difficulty of unlearning. MKSD and MSKSD tends to flag apparent outliers in the training samples as easily unlearnable training samples, whereas EMSKSD flags less obvious but abnormal training samples, such as truck on paper, fake horse or wrongly labeled numbers.

## 5 EVALUATION AND ANALYSIS

In this section, we assess the effectiveness of various unlearning approximation methods on data points ranked from easy to hard for unlearning against the scoring metrics proposed. Specifically, we want to answer the following questions through the experiments:

- Q1: Given easily and hardly unlearnable data points flagged out by the scoring metrics, does the resulting unlearned model show different predictive performance?
- Q2: Which of the proposed scoring metrics show better predictive alignment with actual unlearning outcome?
- Q3: Which unlearning algorithm show better performance when facing hardly unlearnable data points?

### 5.1 EXPERIMENTAL SETUPS

**Datasets and models** We conducted evaluation using three datasets: MNIST(Deng, 2012), CIFAR-10(Krizhevsky et al., 2009), and SVHN (Netzer et al., 2011). We consider a two-layer CNN classifier

Table 1: Average accuracy of the forget set, and test data for the top five easiest (Easy) and five most difficult-to-unlearn samples (Difficult), as recommended by each scoring metric, after unlearning. For the forget set, a lower value shows a better unlearning outcome, whereas, for the test set, a higher value shows a better unlearning outcome. Statistics are given in the Appendix 8.

| Dataset | Method | Forget | | | | | | Test | | | | | |
| | | GradAsct | | FineTune | | Retrain | | GradAsct | | FineTune | | Retrain | |
| | | Easy | Difficult | Easy | Difficult | Easy | Difficult | Easy | Difficult | Easy | Difficult | Easy | Difficult |
| MNIST | PC | 1.00 | 1.00 | 1.00 | 1.00 | 1.00 | 1.00 | 0.99 | 0.99 | 0.99 | 0.99 | 0.99 | 0.99 |
| | MKSD | **0.60** | 1.00 | 1.00 | 1.00 | 1.00 | 1.00 | 0.99 | 0.99 | 0.99 | 0.99 | 0.99 | 0.99 |
| | MSKSD | **0.00** | 1.00 | **0.60** | 1.00 | **0.40** | 1.00 | 0.99 | 0.99 | 0.99 | 0.99 | 0.99 | 0.99 |
| | SSN | **0.40** | 1.00 | **0.40** | 1.00 | 0.60 | 1.00 | 0.99 | 0.99 | 0.99 | 0.99 | 0.99 | 0.99 |
| | EMSKSD | **0.00** | 1.00 | 0.60 | 1.00 | 0.60 | 1.00 | 0.99 | 0.99 | 0.99 | 0.99 | 0.99 | 0.99 |
| CIFAR10 | PC | 0.00 | 0.00 | 1.00 | 1.00 | 1.00 | 1.00 | 0.59 | 0.62 | 0.86 | 0.85 | 0.99 | 0.86 |
| | MKSD | **0.00** | 0.80 | 1.00 | 1.00 | 1.00 | 1.00 | 0.37 | 0.48 | 0.85 | 0.83 | 0.86 | 0.86 |
| | MSKSD | **0.00** | 0.20 | 1.00 | 1.00 | 0.60 | 1.00 | 0.70 | 0.49 | 0.86 | 0.86 | 0.86 | 0.86 |
| | SSN | **0.00** | 0.00 | 0.80 | 1.00 | 0.80 | 1.00 | 0.71 | 0.54 | 0.85 | 0.84 | 0.86 | 0.86 |
| | EMSKSD | **0.00** | 0.00 | 0.80 | 1.00 | **0.60** | 1.00 | **0.73** | **0.53** | 0.85 | 0.83 | 0.86 | 0.86 |
| SVHN | PC | 0.20 | 0.00 | 1.00 | 1.00 | 1.00 | 1.00 | 0.82 | 0.76 | 0.96 | 0.95 | 0.96 | 0.96 |
| | MKSD | **0.00** | 0.20 | 1.00 | 1.00 | 1.00 | 1.00 | 0.45 | 0.74 | 0.96 | 0.96 | 0.96 | 0.96 |
| | MSKSD | **0.00** | 0.00 | 1.00 | 1.00 | 0.40 | 1.00 | 0.81 | 0.75 | 0.96 | 0.96 | 0.96 | 0.96 |
| | SSN | **0.00** | 0.00 | 0.80 | 1.00 | **0.20** | 1.00 | 0.78 | 0.84 | 0.96 | 0.96 | 0.96 | 0.96 |
| | EMSKSD | **0.00** | 0.00 | 0.20 | 1.00 | **0.00** | 1.00 | **0.89** | **0.81** | 0.96 | 0.96 | 0.96 | 0.96 |

for MNIST, and ResNet18 for the CIFAR-10 and SVHN datasets. Details of datasets and models are presented in Appendix Table 4.

**Unlearning methods** We used three primary unlearning baselines i.e. **GradAsc**, **FineTune** , and **Fisher**. Additionally, we retrain the model, a.k.a **Retraining**, without the forgetting set. Although the interest in guaranteed unlearning initially stemmed from the challenges associated with accessing retrained models, achieving guaranteed unlearning for feasible datapoints by retraining presents an intriguing opportunity to explore the characteristics of data points that can be unlearned more readily. If we can achieve guaranteed unlearning for a single data point through retraining, it validates our assumption regarding the varying levels of unlearning feasibility, as it has been shown that generalization ability of ML model (Xu et al., 2024) prevents unlearning single samples.

**Evaluation Criteria** To assess the effectiveness of unlearning, we measured the accuracy of unlearned model on the remaining, forget and test subsets. Additionally, we calculated the layer-wise distance between unlearned and original models. We conducted a MIA on the unlearned model (*MIA-Correctness*) to ascertain how many samples from the forgetting set were correctly classified as non-training samples. We assessed the MIA-efficacy using a confidence-based attack method (Song et al., 2019). The efficacy of MIA is quantified by the ratio of samples predicted as "forgotten samples" (True Negatives *TN*) to the total number of samples in the forgetting set $|\mathcal{D}_f|$. Ideally, post-unlearning, the model $\theta_u$ should have effectively "forgotten" the information related to the samples in forgetting set.

Additional information about experimental setup and hyper-parameter configurations are given in Appendix (see Table 5).

## 5.2 SCORING DIFFICULTY OF UNLEARNING IN TERMS OF PREDICT PERFORMANCE SHIFT

To quantify the effectiveness of the scoring metrics proposed (described in Section 4.2), we selected the top five easiest and the five most difficult samples to unlearn based on the ranking from each of the scoring metrics. In addition, to show the scoring metrics are not unlearning algorithm dependent, we included four well-known unlearning methods to show the alignment of outcomes. Ideally, proposed scoring metrics should identify "easy" and "hard" samples correctly from the training dataset such that "easy" samples are indeed likely to be successfully unlearned, while the "hard" samples may cause either 1) negative impact on models' performance on test set or 2) failure in terms of removing influence of the removal samples.

Table 1 shows the quantitative experimental results, where in addition to the proposed scoring functions, we included "Predictive Confidence" (PC) as a handy baseline scoring method (Liu et al., 2023b; Chen et al., 2023). We note that the most successful unlearning method for ensuring guaranteed unlearning is Grad Ascent (GA) in our experiment. The best scoring metric for detecting both easy and difficult datapoints to be unlearned is EMSKSD, which incorporates both the closeness to

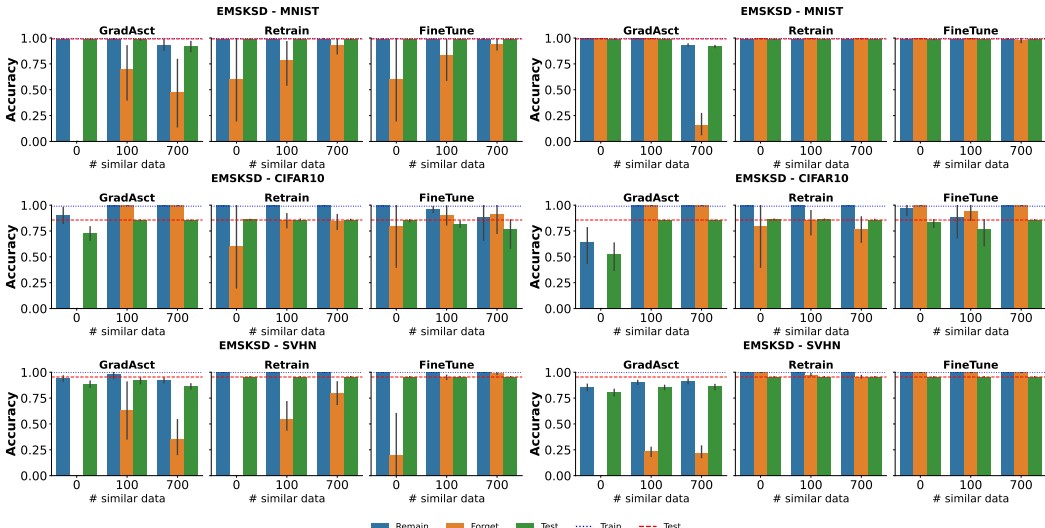

Figure 2: Unlearning accuracy of the remaining, forget, and test subsets for the top 5 easiest and most difficult-to-unlearn datapoints, as recommended by "EMSKSD," including [0, 100, 700] similar samples, evaluated across GradAscent, FineTuning, and Retraining methods.

the decision boundary and the strength of similarity in ranking datapoints. When zooming in the unlearning results on the forget set, it is clear that unlearning the easy datapoints results in large predictive error on them. In contrast, unlearning difficult datapoints either jeopardizes the models' performance or results in unsuccessful unlearning (no predictive difference). When zooming in the unlearning results on the test set, we note there is no distinguishable difference between easy and hard samples in terms of impacting models' generalization, indicating that measuring the models' generalization on a test set will not directly reflecting the effectiveness of unlearning. Indeed, we note the test set can contain data points that are similar to the targeted forget data points such that failing to make correct prediction on them will also reduce the overall accuracy.

## 5.3 SCORING DIFFICULTY OF UNLEARNING IN TERMS OF REMOVAL EXPANSION

To further validate the effectiveness of the difficulty scoring metrics, we conducted experiments to demonstrate the differences between unlearning easy and difficult samples in the context of guaranteed unlearning. Specifically, in these experiments, we extended the unlearning operation to similar samples (based on the Stein kernel value) such that the sample have higher chance to be successfully unlearned. Intuitively, a data sample that requires fewer similar samples (smaller expanded removal set) to be successfully unlearned is considered easier to unlearn, which serves as ground-truth (reflecting feasibility factor described in Section 3: Size of Unlearning Expansion). Figure 2 shows the results; The left-hand side of the figures show the results of "easy" samples ranked by EMSKSD, whereas the right-hand side highlights "difficult" samples ranked by the same scoring metric. From the figure, we observed that the "easy" samples can often be successfully unlearned without much removal set expansion. In contrast, the "difficult" samples are either requiring larger removal set expansion or destroying model's generalization (as shown in green bar). The observation reflects our intuition of unlearning "easy" and "difficult" samples in practice, further showing the scoring metrics proposed providing correct guidance on distinguishing samples' difficulty of unlearning.

A side knowledge we learned from this experiment is that monotonically expanding the removal set is not necessarily helpful in terms of increasing the chance of successful unlearning; the operation may break the model with larger set size more than needed.

## 5.4 SCORING DIFFICULTY OF UNLEARNING IN CONTEXT OF MEMBERSHIP ATTACK

Table 3 presents the MIA-efficacy (Liu et al., 2024b) of unlearning for each of the sorting metrics. The MIA-efficacy reflect the effectiveness of unlearning, where higher MIA-efficiency implies less

Table 2: Average loss of the forget set, and test data for the top five easiest (Easy) and five most difficult-to-unlearn samples (Difficult), as recommended by each scoring metric, after unlearning. For the forget set, a higher value shows a better unlearning outcome, whereas, for the test set, a lower value shows a better unlearning outcome. Statistics are given in the Appendix 9

| Dataset | Method | Forget | | | | | Test | | | | | |
|---|---|---|---|---|---|---|---|---|---|---|---|---|
| | | GradAsct | | FineTune | | Retrain | GradAsct | | FineTune | | Retrain | |
| | | Easy | Difficult | Easy | Difficult | Easy | Easy | Difficult | Easy | Difficult | Easy | Difficult |
| MNIST | PC | 0.00 | 0.00 | 0.00 | 0.00 | 0.00 | 0.03 | 0.03 | 0.00 | 0.04 | 0.03 | 0.03 |
| | MKSD | **1.37** | 0.00 | 0.07 | 0.00 | 0.08 | 0.03 | 0.03 | 0.04 | 0.04 | 0.03 | 0.03 |
| | MSKSD | **4.64** | 0.01 | 1.08 | 0.10 | **1.10** | 0.04 | 0.03 | 0.04 | 0.04 | 0.03 | 0.03 |
| | SSN | **4.73** | 0.00 | 1.68 | 0.00 | **1.09** | 0.04 | 0.03 | 0.04 | 0.04 | 0.03 | 0.03 |
| | EMSKSD | **3.05** | 0.00 | **2.20** | 0.00 | **2.13** | 0.04 | 0.03 | 0.04 | 0.04 | 0.03 | 0.03 |
| CIFAR10 | PC | 5.13 | 5.22 | 0.00 | 0.00 | 0.00 | 5.99 | 5.19 | 0.49 | 0.52 | 0.48 | 0.46 |
| | MKSD | **4.80** | **1.40** | 0.04 | 0.01 | 0.06 | 11.10 | 6.13 | 0.53 | 0.61 | 0.46 | 0.48 |
| | MSKSD | **5.64** | **4.07** | 0.01 | 0.00 | **1.27** | **3.69** | **7.69** | 0.49 | 0.49 | 0.49 | 0.47 |
| | SSN | **5.51** | **5.21** | 0.34 | 0.00 | 0.46 | **3.47** | **6.49** | 0.54 | 0.55 | 0.49 | 0.48 |
| | EMSKSD | **5.48** | **5.79** | **1.10** | 0.00 | 0.57 | **2.89** | **8.06** | 0.53 | 0.55 | 0.48 | 0.47 |
| SVHN | PC | 3.75 | 3.92 | 0.00 | 0.00 | 0.19 | 2.15 | 3.06 | 0.18 | 0.18 | 0.18 | 0.18 |
| | MKSD | **5.12** | **1.71** | 0.00 | 0.00 | 0.00 | 9.36 | 2.81 | 0.17 | 0.18 | 0.18 | 0.18 |
| | MSKSD | **4.51** | **2.33** | 0.05 | 0.00 | **1.53** | 1.74 | 2.39 | 0.18 | 0.17 | 0.18 | 0.18 |
| | SSN | **3.80** | **4.66** | 1.59 | 0.00 | **5.41** | 2.25 | 2.18 | 0.18 | 0.18 | 0.17 | 0.18 |
| | EMSKSD | **3.41** | **4.25** | **2.70** | 0.00 | **11.07** | **1.31** | **2.22** | 0.18 | 0.18 | 0.18 | 0.18 |

Table 3: Average Memebrship Inference Attack Efficacy (MIA-efficacy) for top five easiest and five most difficult to unlearn recommended by each scoring metric. Higher MIA shows better unlearning outcome.

| Dataset | Method | Easy | | | Difficult | | |
|---|---|---|---|---|---|---|---|
| | | GradAsct | FineTune | Retrain | GradAsct | FineTune | Retrain |
| MNIST | PC | 0.00 | 0.00 | 0.00 | 0.00 | 0.00 | 0.00 |
| | MKSD | 0.40 | 0.00 | 0.00 | 0.00 | 0.00 | 0.00 |
| | MSKSD | 1.00 | 0.40 | 0.60 | 0.00 | 0.00 | 0.00 |
| | SSN | 0.60 | 0.60 | 0.40 | 0.00 | 0.00 | 0.00 |
| | EMSKSD | 1.00 | 0.40 | 0.40 | 0.00 | 0.00 | 0.00 |
| CIFAR10 | PC | 1.00 | 0.00 | 0.00 | 1.00 | 0.00 | 0.00 |
| | MKSD | 0.80 | 0.00 | 0.00 | 0.20 | 0.00 | 0.20 |
| | MSKSD | 1.00 | 0.00 | 0.40 | 0.80 | 0.00 | 0.00 |
| | SSN | 1.00 | 0.20 | 0.20 | 1.00 | 0.00 | 0.00 |
| | EMSKSD | 1.00 | 0.20 | 0.40 | 1.00 | 0.00 | 0.20 |
| SVHN | PC | 0.80 | 0.00 | 0.00 | 0.80 | 0.00 | 0.00 |
| | MKSD | 0.00 | 0.00 | 0.20 | 0.80 | 0.00 | 0.00 |
| | MSKSD | 1.00 | 0.00 | 0.60 | 1.00 | 0.00 | 0.00 |
| | SSN | 1.00 | 0.20 | 0.80 | 1.00 | 0.00 | 0.00 |
| | EMSKSD | 1.00 | 0.80 | 1.00 | 1.00 | 0.00 | 0.00 |

information about samples unlearned $D_f$, indicating more successful unlearning outcome. It worth noting that using MIA as evaluation metric in the machine (un)learning literature is still controversial given the subtle metric designs (see (Kong et al., 2023; Murakonda et al., 2021)). We conduct this experiment only to see if there is a rough positive alignment between MIA-efficiency score and the unlearning difficulty score we proposed. From the MIA results, we note that EMSKSD provides better guidance on identifying "easy" and "hard" samples for unlearning in the context of membership attack. Specifically, the "easy" samples identified by EMSKSD consistently show higher MIA-efficacy, whereas "difficulty" samples often with lower MIE-efficacy.

## 5.5 EXPERIMENTAL RESULTS REVISIT

Now, we revisit the research questions that we aimed to answer in our empirical study and evaluate if KSD-based scoring metrics can provide guidance on the difficulty of machine unlearning.

Q1:The accuracy on the forget set reveal that, for certain unlearning algorithms, the model still correctly predicts the class label of unlearned datapoints, despite that the model's error on those particular datapoints has increased. The comparison between the unlearned model's error between difficult and easy datapoints shows a significant gap, indicating that the model struggles more with difficult-to-unlearn samples.

Q2: EMSKSD demonstrates stronger predictive alignment with the actual unlearning outcome, as evidenced by its higher error on the forget set for easy-to-unlearn samples. The key strength of this

sorting heuristic is its ability to combine "predictive confidence" with the "strength of correlation to similar samples" based on the model's estimated distribution. This combination allows EMSKSD to more accurately identify samples that can be effectively unlearned.

Q3: GradAscent proved to be the most effective algorithm for tackling difficult data points, yet it faced challenges in fully dissociating the influence of these points, even when large numbers of similar data points were also unlearned. This method often either compromised the model's performance or failed to significantly improve key unlearning metrics, such as predictive confidence, as shown in Figure 2 and Tables 3. A critical factor is determining the threshold of tolerable performance degradation. If the user can tolerate a certain level of reduction in model performance, it may still be possible to unlearn difficult data points.

### 5.6 Unlearning Feasibility: Insights from Experiments and KSD Scoring

The systematic evaluation of unlearning performance for "easy" and "difficult" samples (recommended by each KSD-based scores 4.2) are tied to the six unlearning difficulty factors 3. For each significant factor affecting the feasibility of unlearning, we establish its relationship with the experimental results.

To evaluate the "Performance Shift", we examined the impact of unlearning on the predictive accuracy and loss of the unlearned models. For an **easy** sample, most unlearning algorithms achieved zero accuracy on the forget set $S_F$ (Figure 2). For **easy** sample the predictive accuracy of the model on the remaining and test subsets remained close to that of the original model (Table 1); and surprisingly unlearning by retraining could achieve $0\%$ accuracy on the forget set. For **difficult** data, the performance of the models was significantly compromised. This was evident from a nearly 50% reduction in accuracy on the test data for CIFAR10. Additionally, unlearning was often unsuccessful, as there was no observable change in the model's accuracy on the forget set, indicating that the unlearning process had no measurable effect. We also have provided "Unlearning Loss" in Table 2. The unlearned models exhibited a significant increase on forget error's which in combination with achieving $0\%$ accuracy on the unlearned sampled, signifies the unlearning was successful and effective for removing the information of "easy" samples from unlearned model. In contrast, the unlearning of "**Difficult**" datapoints had minimal to no impact on the model's error on the forget set, indicating that these datapoints were more resistant to unlearning. Furthermore, the "Resistance to Membership Inference Attack (MIA)" for "easy" samples identified by "**EMSKSD**" consistently show higher MIA-efficacy, whereas "difficult" samples often with lower MIA-efficacy. This observation suggests the algorithm could still identify the unlearned samples as part of the training set, highlighting the failure of the unlearning process for "Difficult" points. The analysis of the rest of the factors is given in Appendix A.4, A.5.

## 6 Conclusion

In this paper, we examined the difficulty of machine unlearning in response to the increasing demand for this process. We summarized the six unlearning feasibility factors that can assess the difficulty of unlearning individual data points, including as size of unlearning expansion, tolerance of performance shift, resistance to membership attack, etc. With the summarized unlearning feasibility factors, we proposed a set of scoring metrics to measure the unlearning difficulty based on Kernelized Stein Discrepancy. Our empirical evaluation shows the proposed scoring metrics can provide guidance on correctly identifying easy and difficult samples ahead of unlearning operation, which is potentially useful in practice for supporting decision making.

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

Table 4: This table details the datasets and models used in evaluating unlearning algorithms, specifying the models applied to each dataset, including the number of layers, batch sizes, number of classes, learning rates, and sample sizes. The information presented provides insight into the computational frameworks employed to analyze MNIST, CIFAR-10, and SVHN datasets, demonstrating the diversity of approaches used in the study.

| Dataset | Model | Layers | Batch Size | Number of Classes | Learning Rate | Samples |
|---------|-------|--------|------------|-------------------|---------------|---------|
| MNIST | 2-layer CNN | 2 | 150 | 10 | 0.001 | 54000 |
| CIFAR10 | ResNet18 | 18 | 150 | 10 | 0.01 | 45000 |
| SVHN | ResNet18 | 18 | 64 | 10 | 0.001 | 58000 |

# A    APPENDIX

## A.1    DATASETS AND UNLEARNING ALGORITHM PARAMETERS

The summary of each model, training parameters and dataset associated for to that model is given in this table.

For the unlearning processes conducted by GradAsct we have introduced a controlling factor a.k.a "Overfitting Threshold" (See Table. 5). The overfitting threshold is a mechanism is only applied on GradAsct during the unlearning process. Without this control, GradAsct can lead to a significant increase in the error rate, rendering the results invalid. To address this issue, we define a cap on the model's error for the forget set, referred to as the "overfitting threshold". This threshold prevents the unlearning loss from becoming excessively high, which mitigates the risk of distorting the unlearned model. In particular, unlearning a single data point with GradAsct requires careful control to ensure the process is effective. By applying this threshold cap, we aim to prevent excessive error growth while preserving the overall quality of the model.

Table 5: Reporting the parameters of each unlearning approximation algorithm for each dataset.

| Dataset | Algorithm | Parameters |
|---------|-----------|------------|
| MNIST | Fine Tuning | LR=0.1, Epochs=10 |
| | Gradient Ascent | overfit_threshold=5.0, LR=1e-4, Epochs=50 |
| | Fisher Forgetting | alpha=1e-5 |
| | Retraining | LR=0.1, Epochs=10 |
| CIFAR10 | Fine Tuning | LR=0.01, Epochs=15 |
| | Gradient Ascent | overfit_threshold=5.0, LR=1e-4, Epochs=50 |
| | Fisher Forgetting | alpha=6e-8 |
| | Retraining | LR=0.01, Epochs=100 |
| SVHN | Fine Tuning | LR=0.01, Epochs=15 |
| | Gradient Ascent | overfit_threshold=5.0, LR=1e-4, Epochs=50 |
| | Fisher Forgetting | alpha=6e-8 |
| | Retraining | LR=0.01, Epochs=100 |

## A.2    COMPUTATION EFFICIENCY

To ensure the computational efficiency of KSD-based scoring metrics, we performed both a detailed analytical evaluation of their computational complexity and experimental assessments through wall-clock time measurements. The analytical evaluation focuses on the time complexity of the KSD score. For each datapoint, metrics such as MKSD, MSKSD, and EMSKSD are computed over all training data $n$ and rely on the Stein Kernel function (Equation 4). The computation of the Stein Kernel is bounded by the term $(k(a, b) \cdot \nabla_a \log P_\theta(a) \cdot \nabla_b \log P_\theta(b)$. A separate analysis of these components reveals the following complexities:

1. The computation of $k(a, b)$ has a constant time complexity of $O(C)$.

2. The computation of $\nabla_a \log P_\theta(a)$ is influenced by the data dimension ($d$) and the network depth ($h$), resulting in a complexity of $O(d \times h)$.

As a result, the Stein Kernel function has a complexity of $O(C \times (d \times h)^2)$. Since the MKSD, MSKSD, and EMSKSD metrics iterate over all data points $(n)$, the overall time complexity of these metrics is bounded by $O(n \times C \times (d \times h)^2)$. For the SSN metric, the complexity is determined by the computation of $\nabla_a P_\theta$, which is bounded by $O(g)$, where $g = O(d \times h)$. Later, to empirically evaluate computational efficiency, we conducted wall-clock time measurements on a subset of the training data $(n' < n)$ and reported the results in Table 6. This empirical assessment provides practical insights into the real-time performance of these metrics.

Table 6: Empirical evaluation of computational efficiency for KSD-based scoring metrics across different datasets and network architectures.

| Dataset | Network | Network Depth (h) | Data Dimension (d) | Instances (n') | Time (n' $\times$ g$^2$) |
|---|---|---|---|---|---|
| MNIST | CNN | 4 | $1 \times 28 \times 28$ | 100 | 33.20 |
| CIFAR10 | ResNet18 | 18 | $3 \times 32 \times 32$ | 128 | 102.81 |
| SVHN | ResNet18 | 18 | $3 \times 32 \times 32$ | 150 | 152.84 |

Additionally the KSD scoring metrics can be efficiently calculated on a personal computer and do not require expensive hardware resources. Details of the system on which we computed all of our scoring metrics are given below. Operating system: Ubuntu 22.04 LTS, Hardware Model: ASUSTeK COMPUTER INC. PRIME Z390-A, CPU: Intel® Core™ i7-9700K × 8, Memory: 16 GB, GPU: NVIDIA TITAN Xp.

### A.3 Easy vs. Difficult Stein Kernel

Investigating the stein kernel similarity help us to understand the difference of stein kernel correlation for a simple versus difficult to unlearn sample. For easy samples, the Stein Kernel reflects the idea of each scoring metric, for example, MKSD averages the stein kernel of each datapoint with the rest of training samples, therefore there are a sharp decline in the stein kernel. MSKSD values the strength of similarity rather the number of similar samples, therefore the most difficult sample has a very strong correlation to small number of training data and then a very sparse and sharp decline in the stein kernel values. SSN has a high similarity to very limited samples and then a large and sharp gap in similarity. Finally the EMSKSD combines the predictive confidence and the similarity strength to rest of data. as the predictive confidence is involved it makes it harder to interpret the Stein Kernels, but it is obvious that the difficult sample has a sharper decline in the Stein Kernel

From comparing the images shown in Figure 1 and the 3, the most difficult samples from CIFAR10 and SVHN selected by every scoring metric have a strong with gradual decline in Stein Kernel Similarity to their corresponding class's data and then a sudden large gap with the rest of the classes, this can help us to understand that the most difficult samples are are strongly similar with the samples within their class, but farther from the other classes. Also this can indicate that both of these datasets, the shape of samples from different classes can be very distinctive. However, For MNIST, even the most difficult the stein kernel similarity reduces but still it includes samples from the same or other classes that can potentially resemble the data. From this figure, the most difficutl to unlearn sample is from class "5". For this sample as the Stein kernel reduces it includes samples from the class "5, 9, 6" which can give us the intuition that these are samples was either closely resembles "5" and could be misclassied.

This comparison also shows the diversity of classes in the most similar samples for an easy vs difficult. Larger diversity in similar sample indicate the higher uncertainly in the prediction of that sample / or closeness to the decision boundaries of other samles / misclassification of that sample. As the SSN and EMSKSD consider consider the deicison boundary in theri ranking, the easies samples shows more diversity in the most similar samples -

-

### A.4 Further Discussion

Among the unlearning algorithms, FineTuning and Retraining are heavily influenced by the remaining data, making the unlearning of a single data point particularly challenging. This challenge is

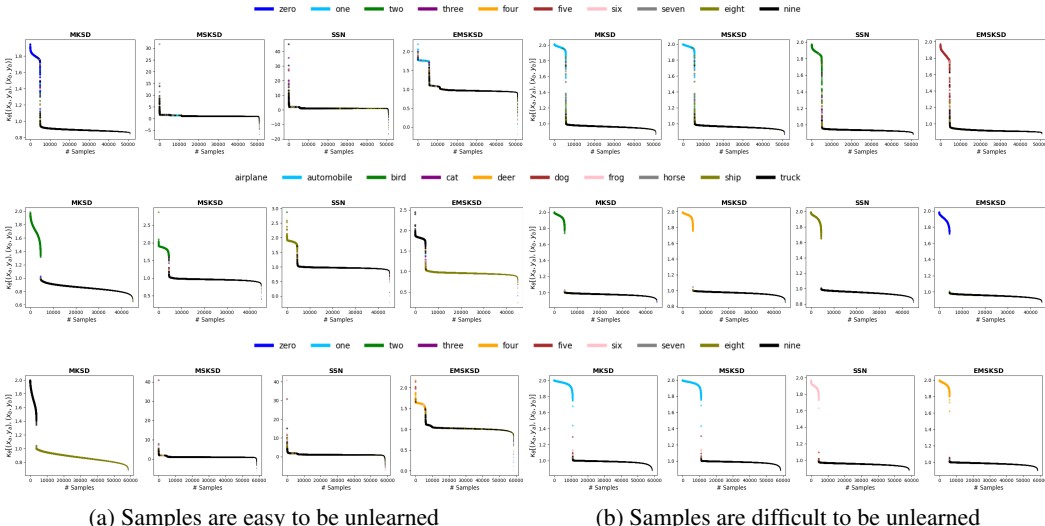

(a) Samples are easy to be unlearned          (b) Samples are difficult to be unlearned

Figure 3: The stein kernel of easiest / most difficult samples recommended by scoring metrics with the rest of training data. The colors indicate the class label of each training data. The stein kernel indicates how strongly each two data are correlated conditioned on the approximated data distribution by model.

especially pronounced in FineTuning, as it requires adjusting the model's parameters and decision boundaries by removing a data point whose influence is still embedded in the model's weights. Significantly shifting the decision boundary to change a sample's class label often requires multiple unlearning steps. In contrast, retraining begins with a model that has no prior knowledge of the data point (particularly those near the decision boundary and less correlated with similar samples), leading to a learned decision boundary that differs from the original model. This distinction can be validated by comparing the error values of the forget set for easily unlearned data points across the FineTune and Retrain algorithms. In nearly all datasets, retraining results in higher error values compared to fine-tuning, further highlighting the greater difficulty of unlearning in the fine-tuning process.

One notable observation is that the accuracy of models in estimating the training data distribution influences the scoring metric and the selection of data points for unlearning. The scoring metric, which is based on the Stein Kernel, is significantly shaped by the model's estimation of the data distribution. For example, the ResNet-18 model achieves 86% predictive accuracy on the CIFAR-10 test set, which is relatively lower compared to models trained on datasets such as SVHN and MNIST, both of which reach 99% training accuracy. Consequently, when the scoring metric suggests data points for unlearning in datasets like CIFAR-10, the model demonstrates better unlearning capacity. This improvement can be attributed to the model's more accurate learning of the data distribution and the decision boundaries between classes

## A.5 LayerWise Distance between Unlearned and Original Models

The layer-wise distance between the original model and the unlearned model is shown in Table 7 . When evaluating the impact of unlearning on model performance, GradAscent demonstrates the smallest distance in relation to the unlearned model, despite being the most effective approach for unlearning. This suggests that although it alters the model's parameters minimally, it still achieves significant unlearning success.

Among the scoring metrics, EMSKSD exhibits the smallest distance on the model's parameters, indicating that it causes less disruption during unlearning. On the other hand, SSN produces the largest gap, as anticipated. This is because SSN tends to select data points near the decision boundary, which typically have larger gradient magnitudes, thereby increasing the distance between the original and unlearned models.

Table 7: Average LayerWise distance between between the original and unlearned model for the top five easiest-to-unlearn and most difficult-to-unlearn samples recommended by each scoring metric.

| Dataset | Sorting Algorithm | Easy | | | | Difficult | | | |
|---|---|---|---|---|---|---|---|---|---|
| | | GradAsct | Fisher | Retrain | FineTune | GradAsct | Fisher | Retrain | FineTune |
| MNIST | PC | 0.00026 | 0.70335 | 0.80291 | 1.03954 | 0.00002 | 0.72223 | 0.80508 | 1.06149 |
| | MKSD | 0.02693 | 0.72219 | 0.80590 | 1.03740 | 0.00013 | 0.70335 | 0.81196 | 1.07146 |
| | MSKSD | 0.00931 | 0.70330 | 0.80968 | 1.04996 | 0.00110 | 0.70335 | 0.80522 | 1.05247 |
| | SSN | 0.01703 | 0.72222 | 0.80066 | 1.07861 | 0.00102 | 0.72015 | 0.80234 | 1.08161 |
| | EMSKSD | 0.00973 | 0.72219 | 0.80945 | 1.05923 | 0.00007 | 0.70333 | 0.81635 | 1.06973 |
| CIFAR10 | PC | 0.01598 | 0.21185 | 3.95690 | 0.39535 | 0.01161 | 0.21377 | 3.76575 | 0.39579 |
| | MKSD | 0.01691 | 0.21433 | 3.86180 | 0.39578 | 0.01532 | 0.21177 | 3.82455 | 0.39600 |
| | MSKSD | 0.01816 | 0.21177 | 3.93015 | 0.39559 | 0.01592 | 0.21378 | 3.64952 | 0.39546 |
| | SSN | 0.01513 | 0.21378 | 3.91489 | 0.39554 | 0.01214 | 0.21185 | 4.19025 | 0.39547 |
| | EMSKSD | 0.01513 | 0.21432 | 3.95492 | 0.39484 | 0.01561 | 0.21176 | 3.88614 | 0.38209 |
| SVHN | PC | 0.00313 | 0.36909 | 5.3614 | 0.51370 | 0.00335 | 0.36857 | 4.40210 | 0.51758 |
| | MKSD | 0.00332 | 0.27947 | 4.56369 | 0.51639 | 0.00298 | 0.28300 | 4.62549 | 0.51768 |
| | MSKSD | 0.00328 | 0.28304 | 4.78339 | 0.51309 | 0.00296 | 0.28299 | 4.43382 | 0.51392 |
| | SSN | 0.00346 | 0.27953 | 4.60636 | 0.51685 | 0.00326 | 0.27947 | 4.22545 | 0.51624 |
| | EMSKSD | 0.00270 | 0.27947 | 4.43430 | 0.50881 | 0.00347 | 0.28299 | 4.41264 | 0.51472 |

## A.6 UNLEARNING ACCURACY

The table presents a comprehensive statistical analysis of unlearning accuracy. It highlights how the unlearning accuracy of the remaining subset is influenced by whether a simple or difficult sample is being unlearned. When comparing the Gradient Ascent (GradAsct) for the easiest samples to unlearn, the remaining accuracy remains close to the model's original performance. However, unlearning more challenging samples significantly compromises the model's performance.

Table 8: Average unlearning accuracy and standard deviation of the forget, and test subsets for the top five easiest and five most difficult-to-unlearn datapoints, as recommended by each scoring metric evaluated across GradAscent, FineTuning, Fisher Forgetting (Fisher), and Retraining methods.

| | | GradAsct | | | FineTune | | | Fisher | | | Retrain | | |
|---|---|---|---|---|---|---|---|---|---|---|---|---|---|
| | | Retain | Forget | Test | Retain | Forget | Test | Retain | Forget | Test | Retain | Forget | Test |
| **Easy** | | | | | | | | | | | | | |
| MNIST (Easy) | MKSD | 1.00 ± 0.00 | 0.60 ± 0.55 | 0.99 ± 0.00 | 0.99 ± 0.00 | 1.00 ± 0.00 | 0.99 ± 0.00 | 0.94 ± 0.02 | 0.60 ± 0.55 | 0.93 ± 0.02 | 0.99 ± 0.00 | 1.00 ± 0.00 | 0.99 ± 0.00 |
| | MSKSD | 1.00 ± 0.00 | 0.00 ± 0.00 | 0.99 ± 0.00 | 0.99 ± 0.00 | 0.60 ± 0.55 | 0.99 ± 0.00 | 0.94 ± 0.02 | 0.40 ± 0.55 | 0.93 ± 0.02 | 0.99 ± 0.00 | 0.40 ± 0.55 | 0.99 ± 0.00 |
| | SSN | 0.99 ± 0.01 | 0.40 ± 0.55 | 0.99 ± 0.01 | 0.99 ± 0.00 | 0.40 ± 0.55 | 0.99 ± 0.00 | 0.94 ± 0.02 | 0.40 ± 0.55 | 0.93 ± 0.02 | 0.99 ± 0.00 | 0.60 ± 0.55 | 0.99 ± 0.00 |
| | EMSKSD | 0.99 ± 0.00 | 0.00 ± 0.00 | 0.99 ± 0.00 | 0.99 ± 0.00 | 0.60 ± 0.55 | 0.99 ± 0.00 | 0.94 ± 0.02 | 0.40 ± 0.55 | 0.93 ± 0.02 | 0.99 ± 0.00 | 0.60 ± 0.55 | 0.99 ± 0.00 |
| | PC | 1.00 ± 0.00 | 1.00 ± 0.00 | 0.99 ± 0.00 | 0.99 ± 0.00 | 1.00 ± 0.00 | 0.99 ± 0.00 | 0.96 ± 0.01 | 1.00 ± 0.00 | 0.95 ± 0.01 | 0.99 ± 0.00 | 1.00 ± 0.00 | 0.99 ± 0.00 |
| CIFAR10 (Easy) | MKSD | 0.43 ± 0.27 | 0.00 ± 0.00 | 0.37 ± 0.22 | 1.00 ± 0.00 | 1.00 ± 0.00 | 0.85 ± 0.02 | 0.97 ± 0.02 | 1.00 ± 0.00 | 0.78 ± 0.02 | 1.00 ± 0.00 | 0.80 ± 0.45 | 0.86 ± 0.00 |
| | MSKSD | 0.88 ± 0.10 | 0.00 ± 0.00 | 0.70 ± 0.07 | 1.00 ± 0.00 | 1.00 ± 0.00 | 0.86 ± 0.00 | 0.97 ± 0.02 | 1.00 ± 0.00 | 0.78 ± 0.02 | 1.00 ± 0.00 | 0.60 ± 0.55 | 0.86 ± 0.01 |
| | SSN | 0.89 ± 0.10 | 0.00 ± 0.00 | 0.71 ± 0.08 | 1.00 ± 0.00 | 0.80 ± 0.45 | 0.85 ± 0.02 | 0.97 ± 0.02 | 1.00 ± 0.00 | 0.78 ± 0.02 | 1.00 ± 0.00 | 0.80 ± 0.45 | 0.86 ± 0.00 |
| | EMSKSD | 0.91 ± 0.11 | 0.00 ± 0.00 | 0.73 ± 0.08 | 1.00 ± 0.00 | 0.80 ± 0.45 | 0.85 ± 0.01 | 0.97 ± 0.02 | 1.00 ± 0.00 | 0.78 ± 0.02 | 1.00 ± 0.00 | 0.60 ± 0.55 | 0.86 ± 0.01 |
| | PC | 0.71 ± 0.12 | 0.00 ± 0.00 | 0.59 ± 0.09 | 1.00 ± 0.00 | 1.00 ± 0.00 | 0.86 ± 0.00 | 0.98 ± 0.01 | 1.00 ± 0.00 | 0.79 ± 0.01 | 1.00 ± 0.00 | 1.00 ± 0.00 | 0.86 ± 0.00 |
| SVHN (Easy) | MKSD | 0.42 ± 0.09 | 0.00 ± 0.00 | 0.45 ± 0.08 | 1.00 ± 0.00 | 1.00 ± 0.00 | 0.96 ± 0.00 | 1.00 ± 0.00 | 1.00 ± 0.00 | 0.95 ± 0.00 | 1.00 ± 0.00 | 0.80 ± 0.45 | 0.96 ± 0.00 |
| | MSKSD | 0.85 ± 0.08 | 0.00 ± 0.00 | 0.81 ± 0.07 | 1.00 ± 0.00 | 1.00 ± 0.00 | 0.96 ± 0.00 | 1.00 ± 0.00 | 1.00 ± 0.00 | 0.95 ± 0.00 | 1.00 ± 0.00 | 0.40 ± 0.55 | 0.96 ± 0.00 |
| | SSN | 0.83 ± 0.13 | 0.00 ± 0.00 | 0.78 ± 0.11 | 1.00 ± 0.00 | 0.80 ± 0.45 | 0.96 ± 0.00 | 1.00 ± 0.00 | 0.80 ± 0.45 | 0.95 ± 0.00 | 1.00 ± 0.00 | 0.20 ± 0.45 | 0.96 ± 0.00 |
| | EMSKSD | 0.94 ± 0.03 | 0.00 ± 0.00 | 0.89 ± 0.04 | 1.00 ± 0.00 | 0.20 ± 0.45 | 0.96 ± 0.00 | 1.00 ± 0.00 | 0.80 ± 0.45 | 0.95 ± 0.00 | 1.00 ± 0.00 | 0.00 ± 0.00 | 0.96 ± 0.00 |
| | PC | 0.86 ± 0.04 | 0.20 ± 0.45 | 0.82 ± 0.04 | 1.00 ± 0.00 | 1.00 ± 0.00 | 0.96 ± 0.00 | 1.00 ± 0.00 | 1.00 ± 0.00 | 0.95 ± 0.00 | 1.00 ± 0.00 | 1.00 ± 0.00 | 0.95 ± 0.00 |
| **Difficult** | | | | | | | | | | | | | |
| MNIST (Difficult) | MKSD | 1.00 ± 0.00 | 1.00 ± 0.00 | 0.99 ± 0.00 | 0.99 ± 0.00 | 1.00 ± 0.00 | 0.99 ± 0.00 | 0.94 ± 0.02 | 1.00 ± 0.00 | 0.93 ± 0.02 | 0.99 ± 0.00 | 1.00 ± 0.00 | 0.99 ± 0.00 |
| | MSKSD | 1.00 ± 0.00 | 1.00 ± 0.00 | 0.99 ± 0.00 | 0.99 ± 0.00 | 1.00 ± 0.00 | 0.99 ± 0.00 | 0.94 ± 0.02 | 1.00 ± 0.00 | 0.93 ± 0.02 | 0.99 ± 0.00 | 1.00 ± 0.00 | 0.99 ± 0.00 |
| | SSN | 1.00 ± 0.00 | 1.00 ± 0.00 | 0.99 ± 0.00 | 0.99 ± 0.00 | 1.00 ± 0.00 | 0.99 ± 0.00 | 0.94 ± 0.02 | 1.00 ± 0.00 | 0.93 ± 0.02 | 0.99 ± 0.00 | 1.00 ± 0.00 | 0.99 ± 0.00 |
| | EMSKSD | 1.00 ± 0.00 | 1.00 ± 0.00 | 0.99 ± 0.00 | 0.99 ± 0.00 | 1.00 ± 0.00 | 0.99 ± 0.00 | 0.94 ± 0.02 | 1.00 ± 0.00 | 0.93 ± 0.02 | 0.99 ± 0.00 | 1.00 ± 0.00 | 0.99 ± 0.00 |
| | PC | 1.00 ± 0.00 | 1.00 ± 0.00 | 0.99 ± 0.00 | 0.99 ± 0.00 | 1.00 ± 0.00 | 0.99 ± 0.00 | 0.96 ± 0.01 | 1.00 ± 0.00 | 0.95 ± 0.01 | 0.99 ± 0.00 | 1.00 ± 0.00 | 0.99 ± 0.00 |
| CIFAR10 (Difficult) | MKSD | 0.58 ± 0.11 | 1.00 ± 0.00 | 0.48 ± 0.08 | 0.98 ± 0.03 | 1.00 ± 0.00 | 0.83 ± 0.03 | 0.97 ± 0.02 | 1.00 ± 0.00 | 0.78 ± 0.02 | 1.00 ± 0.00 | 0.80 ± 0.45 | 0.86 ± 0.00 |
| | MSKSD | 0.58 ± 0.10 | 0.20 ± 0.45 | 0.49 ± 0.08 | 1.00 ± 0.00 | 1.00 ± 0.00 | 0.86 ± 0.00 | 0.97 ± 0.02 | 0.80 ± 0.45 | 0.78 ± 0.02 | 1.00 ± 0.00 | 1.00 ± 0.00 | 0.86 ± 0.00 |
| | SSN | 0.68 ± 0.14 | 0.00 ± 0.00 | 0.54 ± 0.10 | 0.99 ± 0.02 | 1.00 ± 0.00 | 0.84 ± 0.03 | 0.97 ± 0.02 | 1.00 ± 0.00 | 0.78 ± 0.02 | 1.00 ± 0.00 | 1.00 ± 0.00 | 0.86 ± 0.01 |
| | EMSKSD | 0.64 ± 0.24 | 0.00 ± 0.00 | 0.53 ± 0.18 | 0.97 ± 0.07 | 1.00 ± 0.00 | 0.83 ± 0.05 | 0.97 ± 0.02 | 1.00 ± 0.00 | 0.78 ± 0.02 | 1.00 ± 0.00 | 0.80 ± 0.45 | 0.86 ± 0.00 |
| | PC | 0.77 ± 0.10 | 0.00 ± 0.00 | 0.62 ± 0.08 | 1.00 ± 0.00 | 1.00 ± 0.00 | 0.85 ± 0.01 | 0.98 ± 0.01 | 1.00 ± 0.00 | 0.79 ± 0.01 | 1.00 ± 0.00 | 1.00 ± 0.00 | 0.86 ± 0.00 |
| SVHN (Difficult) | MKSD | 0.79 ± 0.07 | 0.20 ± 0.45 | 0.74 ± 0.07 | 1.00 ± 0.00 | 1.00 ± 0.00 | 0.96 ± 0.00 | 1.00 ± 0.00 | 1.00 ± 0.00 | 0.95 ± 0.00 | 1.00 ± 0.00 | 1.00 ± 0.00 | 0.96 ± 0.00 |
| | MSKSD | 0.81 ± 0.10 | 0.00 ± 0.00 | 0.75 ± 0.10 | 1.00 ± 0.00 | 1.00 ± 0.00 | 0.96 ± 0.00 | 1.00 ± 0.00 | 1.00 ± 0.00 | 0.95 ± 0.00 | 1.00 ± 0.00 | 1.00 ± 0.00 | 0.96 ± 0.00 |
| | SSN | 0.88 ± 0.03 | 0.00 ± 0.00 | 0.84 ± 0.04 | 1.00 ± 0.00 | 1.00 ± 0.00 | 0.96 ± 0.00 | 1.00 ± 0.00 | 1.00 ± 0.00 | 0.95 ± 0.00 | 1.00 ± 0.00 | 1.00 ± 0.00 | 0.96 ± 0.00 |
| | EMSKSD | 0.86 ± 0.04 | 0.00 ± 0.00 | 0.81 ± 0.04 | 1.00 ± 0.00 | 1.00 ± 0.00 | 0.96 ± 0.00 | 1.00 ± 0.00 | 1.00 ± 0.00 | 0.95 ± 0.00 | 1.00 ± 0.00 | 1.00 ± 0.00 | 0.96 ± 0.00 |
| | PC | 0.79 ± 0.10 | 0.00 ± 0.00 | 0.76 ± 0.09 | 1.00 ± 0.00 | 1.00 ± 0.00 | 0.95 ± 0.00 | 1.00 ± 0.00 | 1.00 ± 0.00 | 0.95 ± 0.00 | 1.00 ± 0.00 | 1.00 ± 0.00 | 0.96 ± 0.00 |

## A.7 UNLEARNING LOSS

The unlearned models loss on the forget and test subsets are presented in Table 9), where EMSKSD consistently identifies datapoints with the highest unlearning error. A noteworthy observation from comparing accuracy and loss is that for the easy-to-unlearn cases, the unlearning process appears to have minimal impact on the model's accuracy on the forget set 8. However, including unlearned models' loss analysis of the loss table reveals that, despite the sustained accuracy, the unlearning process increases the models' loss on the forget set for easy-to-unlearn datapoints. This suggests that while the model can still predict the forget datapoints, the error threshold for flipping a datapoint's label remains high. In contrast, for difficult-to-unlearn samples, unlearning has minimal to no impact on the model's error on the forget set, indicating that these datapoints are more resistant to unlearning.

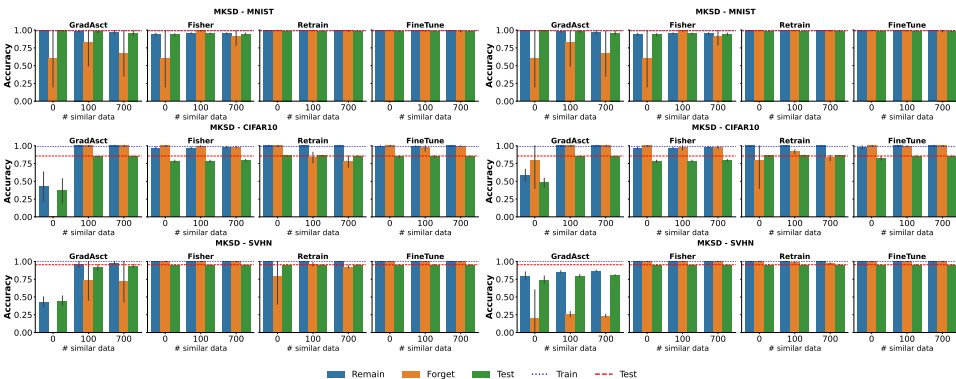

Figure 4: Averaged unlearning accuracy of the remaining, forget, and test subsets for the top five easiest and five most difficult-to-unlearn datapoints, as recommended by "MKSD," including [0, 100, 700] similar samples, evaluated across GradAscent, FineTuning, Fisher Forgetting (Fisher), and Retraining methods.

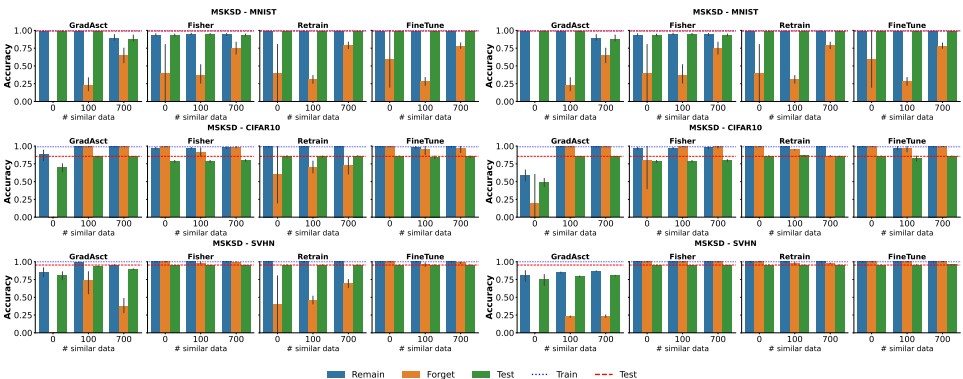

Figure 5: Averaged unlearning accuracy of the remaining, forget, and test subsets for the top 5 easiest and most difficult-to-unlearn datapoints, as recommended by "MSKSD," including [0, 100, 700] similar samples, evaluated across GradAscent, FineTuning, Fisher Forgetting (Fisher), and Retraining methods.

## A.8 MEMBERSHIP INFERENCE ATTACK

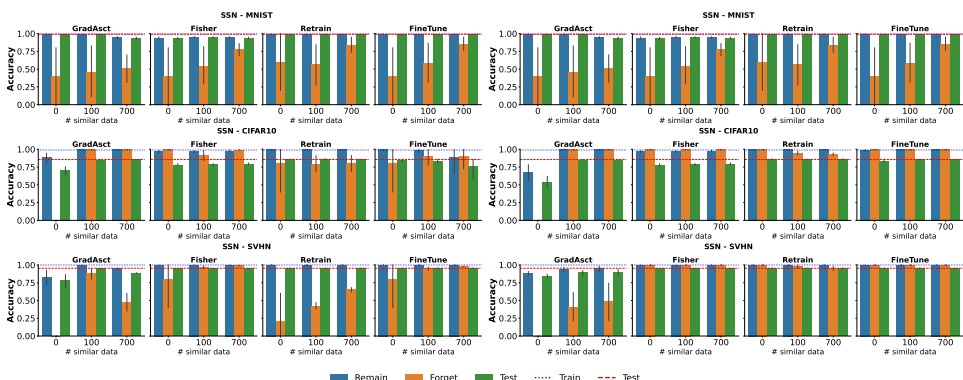

Figure 6: Averaged unlearning accuracy of the remaining, forget, and test subsets for the top 5 easiest and most difficult-to-unlearn datapoints, as recommended by "SSN," including [0, 100, 700] similar samples, evaluated across GradAscent, FineTuning, Fisher Forgetting (Fisher), and Retraining methods.

Table 9: Averaged unlearned model loss of the forget, and test subsets for the top five easiest and five most difficult-to-unlearn datapoints, as recommended by each scoring metric evaluated across GradAscent, FineTuning, Fisher Forgetting (Fisher), and Retraining methods.

| | | GradAsct Retain | GradAsct Forget | GradAsct Test | FineTune Retain | FineTune Forget | FineTune Test | Fisher Retain | Fisher Forget | Fisher Test | Retrain Retain | Retrain Forget | Retrain Test |
|---|---|---|---|---|---|---|---|---|---|---|---|---|---|
| **Easy** | | | | | | | | | | | | | |
| Blob | MKSD | 0.03 ± 0.01 | 1.44 ± 0.39 | 0.04 ± 0.01 | 0.01 ± 0.00 | 0.29 ± 0.12 | 0.01 ± 0.00 | 0.09 ± 0.05 | 1.20 ± 0.87 | 0.08 ± 0.05 | 0.01 ± 0.00 | 0.37 ± 0.14 | 0.01 ± 0.00 |
| | MSKSD | 0.02 ± 0.00 | 1.34 ± 0.22 | 0.02 ± 0.00 | 0.01 ± 0.00 | 0.59 ± 0.07 | 0.01 ± 0.00 | 0.08 ± 0.05 | 1.24 ± 0.76 | 0.08 ± 0.05 | 0.01 ± 0.00 | 0.70 ± 0.11 | 0.01 ± 0.00 |
| | SSN | 0.02 ± 0.00 | 1.34 ± 0.22 | 0.02 ± 0.00 | 0.01 ± 0.00 | 0.59 ± 0.07 | 0.01 ± 0.00 | 0.08 ± 0.05 | 1.24 ± 0.76 | 0.08 ± 0.05 | 0.01 ± 0.00 | 0.70 ± 0.11 | 0.01 ± 0.00 |
| | EMSKSD | 0.02 ± 0.01 | 1.36 ± 0.33 | 0.03 ± 0.01 | 0.01 ± 0.00 | 0.44 ± 0.10 | 0.01 ± 0.00 | 0.09 ± 0.05 | 1.20 ± 0.67 | 0.08 ± 0.05 | 0.01 ± 0.00 | 0.53 ± 0.11 | 0.01 ± 0.00 |
| | PC | 0.02 ± 0.01 | 1.36 ± 0.33 | 0.03 ± 0.01 | 0.01 ± 0.00 | 0.44 ± 0.08 | 0.01 ± 0.00 | 0.08 ± 0.06 | 1.69 ± 2.24 | 0.08 ± 0.07 | 0.01 ± 0.00 | 0.50 ± 0.09 | 0.01 ± 0.00 |
| MNIST | MKSD | 0.01 ± 0.00 | 1.37 ± 1.92 | 0.03 ± 0.00 | 0.02 ± 0.01 | 0.07 ± 0.16 | 0.04 ± 0.01 | 0.21 ± 0.07 | 1.37 ± 1.89 | 0.23 ± 0.06 | 0.02 ± 0.00 | 0.08 ± 0.14 | 0.03 ± 0.00 |
| | MSKSD | 0.02 ± 0.00 | 4.64 ± 2.32 | 0.04 ± 0.01 | 0.03 ± 0.00 | 1.08 ± 0.97 | 0.04 ± 0.01 | 0.21 ± 0.07 | 5.95 ± 6.77 | 0.23 ± 0.06 | 0.02 ± 0.00 | 1.10 ± 1.00 | 0.03 ± 0.00 |
| | SSN | 0.02 ± 0.02 | 4.73 ± 4.84 | 0.04 ± 0.02 | 0.02 ± 0.00 | 1.68 ± 1.69 | 0.04 ± 0.00 | 0.21 ± 0.07 | 3.60 ± 3.44 | 0.23 ± 0.06 | 0.02 ± 0.00 | 1.09 ± 1.45 | 0.03 ± 0.00 |
| | EMSKSD | 0.02 ± 0.01 | 3.05 ± 1.09 | 0.04 ± 0.01 | 0.03 ± 0.00 | 2.20 ± 3.26 | 0.04 ± 0.00 | 0.21 ± 0.07 | 2.77 ± 4.27 | 0.23 ± 0.06 | 0.02 ± 0.00 | 2.13 ± 2.98 | 0.03 ± 0.00 |
| | PC | 0.01 ± 0.00 | 0.00 ± 0.00 | 0.03 ± 0.00 | 0.00 ± 0.00 | 0.00 ± 0.00 | 0.04 ± 0.00 | 0.16 ± 0.06 | 0.00 ± 0.00 | 0.19 ± 0.06 | 0.02 ± 0.00 | 0.00 ± 0.00 | 0.03 ± 0.00 |
| CIFAR10 | MKSD | 9.96 ± 8.52 | 4.80 ± 1.92 | 11.10 ± 7.79 | 0.01 ± 0.02 | 0.04 ± 0.08 | 0.53 ± 0.10 | 0.15 ± 0.04 | 0.02 ± 0.02 | 0.68 ± 0.07 | 0.00 ± 0.00 | 0.06 ± 0.11 | 0.46 ± 0.01 |
| | MSKSD | 1.08 ± 1.22 | 5.64 ± 0.19 | 3.69 ± 1.57 | 0.00 ± 0.00 | 0.01 ± 0.01 | 0.49 ± 0.00 | 0.15 ± 0.04 | 0.29 ± 0.26 | 0.68 ± 0.07 | 0.00 ± 0.00 | 1.27 ± 1.42 | 0.49 ± 0.03 |
| | SSN | 0.98 ± 1.26 | 5.51 ± 0.24 | 3.47 ± 1.60 | 0.00 ± 0.01 | 0.34 ± 0.76 | 0.54 ± 0.12 | 0.15 ± 0.04 | 0.23 ± 0.14 | 0.68 ± 0.07 | 0.00 ± 0.00 | 0.46 ± 0.57 | 0.49 ± 0.04 |
| | EMSKSD | 0.75 ± 1.08 | 5.48 ± 0.28 | 2.89 ± 1.37 | 0.00 ± 0.01 | 1.10 ± 2.45 | 0.53 ± 0.08 | 0.15 ± 0.04 | 0.12 ± 0.06 | 0.68 ± 0.07 | 0.00 ± 0.00 | 0.57 ± 0.80 | 0.48 ± 0.04 |
| | PC | 3.38 ± 2.09 | 5.13 ± 1.89 | 5.99 ± 2.10 | 0.00 ± 0.00 | 0.00 ± 0.00 | 0.49 ± 0.00 | 0.13 ± 0.02 | 0.04 ± 0.04 | 0.66 ± 0.02 | 0.00 ± 0.00 | 0.00 ± 0.00 | 0.48 ± 0.02 |
| SVHN | MKSD | 10.29 ± 1.96 | 5.12 ± 0.14 | 9.36 ± 1.77 | 0.00 ± 0.00 | 0.00 ± 0.01 | 0.17 ± 0.00 | 0.02 ± 0.00 | 0.02 ± 0.02 | 0.19 ± 0.00 | 0.00 ± 0.00 | 0.20 ± 0.44 | 0.18 ± 0.00 |
| | MSKSD | 1.04 ± 0.81 | 4.51 ± 0.32 | 1.74 ± 0.84 | 0.00 ± 0.00 | 0.05 ± 0.05 | 0.18 ± 0.01 | 0.02 ± 0.00 | 0.17 ± 0.14 | 0.19 ± 0.00 | 0.00 ± 0.00 | 1.55 ± 1.51 | 0.18 ± 0.00 |
| | SSN | 1.53 ± 1.40 | 3.80 ± 0.79 | 2.25 ± 1.30 | 0.00 ± 0.00 | 1.59 ± 3.20 | 0.18 ± 0.00 | 0.02 ± 0.00 | 0.76 ± 1.08 | 0.19 ± 0.00 | 0.00 ± 0.00 | 5.63 ± 6.38 | 0.18 ± 0.00 |
| | EMSKSD | 0.42 ± 0.17 | 3.41 ± 0.61 | 1.31 ± 0.23 | 0.00 ± 0.00 | 2.70 ± 2.64 | 0.18 ± 0.01 | 0.02 ± 0.00 | 1.12 ± 0.75 | 0.19 ± 0.00 | 0.00 ± 0.00 | 11.14 ± 1.85 | 0.18 ± 0.00 |
| | PC | 1.41 ± 0.52 | 3.75 ± 1.24 | 2.15 ± 0.55 | 0.00 ± 0.00 | 0.00 ± 0.00 | 0.49 ± 0.00 | 0.02 ± 0.00 | 0.00 ± 0.00 | 0.19 ± 0.00 | 0.00 ± 0.00 | 0.00 ± 0.00 | 0.18 ± 0.00 |
| **Difficult** | | | | | | | | | | | | | |
| Blob | MKSD | 0.01 ± 0.00 | 0.00 ± 0.00 | 0.01 ± 0.00 | 0.01 ± 0.00 | 0.00 ± 0.00 | 0.01 ± 0.00 | 0.09 ± 0.04 | 0.02 ± 0.03 | 0.08 ± 0.04 | 0.01 ± 0.00 | 0.00 ± 0.00 | 0.01 ± 0.00 |
| | MSKSD | 0.01 ± 0.00 | 0.00 ± 0.00 | 0.01 ± 0.00 | 0.01 ± 0.00 | 0.00 ± 0.00 | 0.01 ± 0.00 | 0.09 ± 0.04 | 0.02 ± 0.03 | 0.08 ± 0.04 | 0.01 ± 0.00 | 0.00 ± 0.00 | 0.01 ± 0.00 |
| | SSN | 0.01 ± 0.00 | 0.00 ± 0.00 | 0.01 ± 0.00 | 0.01 ± 0.00 | 0.00 ± 0.00 | 0.01 ± 0.00 | 0.09 ± 0.04 | 0.00 ± 0.00 | 0.08 ± 0.04 | 0.01 ± 0.00 | 0.00 ± 0.00 | 0.01 ± 0.00 |
| | EMSKSD | 0.01 ± 0.00 | 0.00 ± 0.00 | 0.01 ± 0.00 | 0.01 ± 0.00 | 0.00 ± 0.00 | 0.01 ± 0.00 | 0.09 ± 0.04 | 0.00 ± 0.00 | 0.08 ± 0.04 | 0.01 ± 0.00 | 0.00 ± 0.00 | 0.01 ± 0.00 |
| | PC | 0.01 ± 0.00 | 0.00 ± 0.00 | 0.01 ± 0.00 | 0.01 ± 0.00 | 0.00 ± 0.00 | 0.01 ± 0.00 | 0.08 ± 0.06 | 0.01 ± 0.02 | 0.08 ± 0.07 | 0.01 ± 0.00 | 0.00 ± 0.00 | 0.01 ± 0.00 |
| MNIST | MKSD | 0.01 ± 0.00 | 0.00 ± 0.00 | 0.03 ± 0.00 | 0.02 ± 0.00 | 0.00 ± 0.00 | 0.04 ± 0.00 | 0.21 ± 0.07 | 0.03 ± 0.05 | 0.23 ± 0.06 | 0.02 ± 0.00 | 0.00 ± 0.00 | 0.03 ± 0.00 |
| | MSKSD | 0.01 ± 0.00 | 0.01 ± 0.01 | 0.03 ± 0.00 | 0.02 ± 0.00 | 0.10 ± 0.17 | 0.04 ± 0.00 | 0.21 ± 0.07 | 0.24 ± 0.33 | 0.23 ± 0.06 | 0.02 ± 0.00 | 0.02 ± 0.02 | 0.03 ± 0.00 |
| | SSN | 0.01 ± 0.00 | 0.00 ± 0.00 | 0.03 ± 0.00 | 0.02 ± 0.00 | 0.00 ± 0.00 | 0.04 ± 0.00 | 0.21 ± 0.07 | 0.00 ± 0.00 | 0.23 ± 0.06 | 0.02 ± 0.00 | 0.00 ± 0.00 | 0.03 ± 0.00 |
| | EMSKSD | 0.01 ± 0.00 | 0.00 ± 0.00 | 0.03 ± 0.00 | 0.02 ± 0.00 | 0.00 ± 0.00 | 0.04 ± 0.01 | 0.21 ± 0.07 | 0.00 ± 0.00 | 0.23 ± 0.06 | 0.02 ± 0.00 | 0.00 ± 0.00 | 0.03 ± 0.00 |
| | PC | 0.01 ± 0.00 | 0.00 ± 0.00 | 0.03 ± 0.00 | 0.03 ± 0.00 | 0.00 ± 0.00 | 0.04 ± 0.00 | 0.16 ± 0.06 | 0.00 ± 0.00 | 0.19 ± 0.06 | 0.02 ± 0.00 | 0.00 ± 0.00 | 0.03 ± 0.00 |
| CIFAR10 | MKSD | 4.32 ± 1.79 | 1.40 ± 2.86 | 6.13 ± 1.59 | 0.06 ± 0.09 | 0.01 ± 0.01 | 0.61 ± 0.12 | 0.15 ± 0.04 | 0.05 ± 0.05 | 0.68 ± 0.07 | 0.00 ± 0.00 | 0.79 ± 1.70 | 0.48 ± 0.02 |
| | MSKSD | 5.20 ± 2.09 | 4.07 ± 2.44 | 7.69 ± 1.85 | 0.00 ± 0.00 | 0.00 ± 0.00 | 0.49 ± 0.00 | 0.15 ± 0.04 | 0.28 ± 0.60 | 0.68 ± 0.07 | 0.00 ± 0.00 | 0.02 ± 0.04 | 0.47 ± 0.01 |
| | SSN | 4.03 ± 2.12 | 5.21 ± 2.07 | 6.49 ± 1.99 | 0.04 ± 0.06 | 0.00 ± 0.00 | 0.55 ± 0.09 | 0.15 ± 0.04 | 0.02 ± 0.04 | 0.68 ± 0.07 | 0.00 ± 0.00 | 0.00 ± 0.00 | 0.48 ± 0.03 |
| | EMSKSD | 5.71 ± 5.27 | 5.79 ± 0.76 | 8.06 ± 4.69 | 0.10 ± 0.21 | 0.00 ± 0.00 | 0.55 ± 0.15 | 0.15 ± 0.04 | 0.00 ± 0.00 | 0.68 ± 0.07 | 0.00 ± 0.00 | 0.36 ± 0.78 | 0.47 ± 0.01 |
| | PC | 2.44 ± 1.64 | 5.22 ± 1.28 | 5.19 ± 2.06 | 0.00 ± 0.00 | 0.00 ± 0.00 | 0.52 ± 0.06 | 0.13 ± 0.02 | 0.04 ± 0.08 | 0.66 ± 0.02 | 0.00 ± 0.00 | 0.02 ± 0.04 | 0.46 ± 0.01 |
| SVHN | MKSD | 1.91 ± 0.89 | 1.71 ± 0.40 | 2.81 ± 1.16 | 0.00 ± 0.00 | 0.00 ± 0.00 | 0.18 ± 0.00 | 0.02 ± 0.00 | 0.01 ± 0.02 | 0.19 ± 0.00 | 0.00 ± 0.00 | 0.00 ± 0.01 | 0.18 ± 0.00 |
| | MSKSD | 1.49 ± 1.10 | 2.33 ± 0.31 | 2.39 ± 1.21 | 0.00 ± 0.00 | 0.00 ± 0.00 | 0.17 ± 0.00 | 0.02 ± 0.00 | 0.01 ± 0.01 | 0.19 ± 0.00 | 0.00 ± 0.00 | 0.07 ± 0.08 | 0.18 ± 0.00 |
| | SSN | 1.41 ± 0.38 | 4.66 ± 0.70 | 2.18 ± 0.46 | 0.00 ± 0.00 | 0.00 ± 0.00 | 0.18 ± 0.01 | 0.02 ± 0.00 | 0.00 ± 0.00 | 0.19 ± 0.00 | 0.00 ± 0.00 | 0.00 ± 0.00 | 0.18 ± 0.00 |
| | EMSKSD | 1.53 ± 0.57 | 4.25 ± 0.86 | 2.22 ± 0.66 | 0.00 ± 0.00 | 0.00 ± 0.00 | 0.18 ± 0.01 | 0.02 ± 0.00 | 0.00 ± 0.00 | 0.19 ± 0.00 | 0.00 ± 0.00 | 0.00 ± 0.00 | 0.18 ± 0.00 |
| | PC | 2.39 ± 1.74 | 3.92 ± 1.04 | 3.06 ± 1.46 | 0.00 ± 0.00 | 0.00 ± 0.00 | 0.18 ± 0.01 | 0.02 ± 0.00 | 0.00 ± 0.00 | 0.19 ± 0.00 | 0.00 ± 0.00 | 0.00 ± 0.00 | 0.18 ± 0.01 |

Table 10: Average Memebrship Inference Attack Efficacy (MIA-efficacy) for top five easiest and five most difficult to unlearn recommended by each scoring metric. Higher MIA shows better unlearning outcome.

| Dataset | Method | Easy GradAsct | Easy FineTune | Easy Fisher | Easy Retrain | Difficult GradAsct | Difficult FineTune | Difficult Fisher | Difficult Retrain |
|---|---|---|---|---|---|---|---|---|---|
| MNIST | MKSD | 0.40 ± 0.55 | 0.00 ± 0.00 | 0.20 ± 0.45 | 0.00 ± 0.00 | 0.00 ± 0.00 | 0.00 ± 0.00 | 0.20 ± 0.45 | 0.00 ± 0.00 |
| | MSKSD | 1.00 ± 0.00 | 0.40 ± 0.55 | 0.40 ± 0.55 | 0.60 ± 0.55 | 0.00 ± 0.00 | 0.00 ± 0.00 | 0.20 ± 0.45 | 0.00 ± 0.00 |
| | SSN | 0.60 ± 0.55 | 0.60 ± 0.55 | 0.40 ± 0.55 | 0.40 ± 0.55 | 0.00 ± 0.00 | 0.00 ± 0.00 | 0.20 ± 0.45 | 0.00 ± 0.00 |
| | EMSKSD | 1.00 ± 0.00 | 0.40 ± 0.55 | 0.40 ± 0.55 | 0.40 ± 0.55 | 0.00 ± 0.00 | 0.00 ± 0.00 | 0.20 ± 0.45 | 0.00 ± 0.00 |
| | PC | 0.00 ± 0.00 | 0.00 ± 0.00 | 0.20 ± 0.45 | 0.00 ± 0.00 | 0.00 ± 0.00 | 0.00 ± 0.00 | 0.20 ± 0.45 | 0.00 ± 0.00 |
| CIFAR10 | MKSD | 0.80 ± 0.45 | 0.00 ± 0.00 | 0.00 ± 0.00 | 0.00 ± 0.00 | 0.20 ± 0.45 | 0.00 ± 0.00 | 0.00 ± 0.00 | 0.20 ± 0.45 |
| | MSKSD | 1.00 ± 0.00 | 0.00 ± 0.00 | 0.00 ± 0.00 | 0.40 ± 0.55 | 0.80 ± 0.45 | 0.00 ± 0.00 | 0.20 ± 0.45 | 0.00 ± 0.00 |
| | SSN | 1.00 ± 0.00 | 0.20 ± 0.45 | 0.00 ± 0.00 | 0.20 ± 0.45 | 1.00 ± 0.00 | 0.00 ± 0.00 | 0.00 ± 0.00 | 0.00 ± 0.00 |
| | EMSKSD | 1.00 ± 0.00 | 0.20 ± 0.45 | 0.00 ± 0.00 | 0.40 ± 0.55 | 1.00 ± 0.00 | 0.00 ± 0.00 | 0.00 ± 0.00 | 0.20 ± 0.45 |
| | PC | 1.00 ± 0.00 | 0.00 ± 0.00 | 0.00 ± 0.00 | 0.00 ± 0.00 | 1.00 ± 0.00 | 0.00 ± 0.00 | 0.00 ± 0.00 | 0.00 ± 0.00 |
| SVHN | MKSD | 0.00 ± 0.00 | 0.00 ± 0.00 | 0.00 ± 0.00 | 0.20 ± 0.45 | 0.80 ± 0.45 | 0.00 ± 0.00 | 0.00 ± 0.00 | 0.00 ± 0.00 |
| | MSKSD | 1.00 ± 0.00 | 0.00 ± 0.00 | 0.00 ± 0.00 | 0.60 ± 0.55 | 1.00 ± 0.00 | 0.00 ± 0.00 | 0.00 ± 0.00 | 0.00 ± 0.00 |
| | SSN | 1.00 ± 0.00 | 0.20 ± 0.45 | 0.20 ± 0.45 | 0.80 ± 0.45 | 1.00 ± 0.00 | 0.00 ± 0.00 | 0.00 ± 0.00 | 0.00 ± 0.00 |
| | EMSKSD | 1.00 ± 0.00 | 0.80 ± 0.45 | 0.20 ± 0.45 | 1.00 ± 0.00 | 1.00 ± 0.00 | 0.00 ± 0.00 | 0.00 ± 0.00 | 0.00 ± 0.00 |
| | PC | 0.80 ± 0.45 | 0.00 ± 0.00 | 0.00 ± 0.00 | 0.00 ± 0.00 | 0.80 ± 0.45 | 0.00 ± 0.00 | 0.00 ± 0.00 | 0.00 ± 0.00 |

