# OpenReview forum: "Towards Understanding the Feasibility of Machine Unlearning"
_ICLR.cc/2025/Conference — Submitted to ICLR 2025_

### Official Review · Reviewer_V6u3 · 2024-10-31

**Soundness:** 2
**Presentation:** 3
**Contribution:** 2
**Rating:** 6
**Confidence:** 4

**Summary:**

This paper tackles the problem of assessing the difficulty of unlearning individual training samples in machine learning models, a need highlighted by recent privacy regulations. While most existing unlearning methods focus on overall unlearning success rates, this work shifts attention to the unique challenges of unlearning specific samples, considering factors like the underlying model and data characteristics. The authors propose heuristics to predict the success of unlearning operations for individual data points and explore variations in unlearning difficulty across samples, with a ranking mechanism to identify samples that are more resistant to unlearning. A key contribution is the use of Kernelized Stein Discrepancy (KSD) as a model- and data-specific heuristic for gauging unlearning difficulty. The method’s effectiveness is demonstrated on multiple classification tasks, showcasing its applicability across diverse scenarios and highlighting its potential to refine the measurement of unlearning success at a granular level.

**Strengths:**

* This work introduces an original and timely contribution to the field of unlearning by tackling a previously overlooked question: the unlearnability of specific samples.
* The use of Kernelized Stein Discrepancy (KSD) in this context is both innovative and technically sound. The KSD-based unlearnability score, which incorporates model and data characteristics, is compelling.

**Weaknesses:**

* **Regulatory Implications.** It is not clear that the developed tools are useful for advancing unlearning techniques to comply with regulations. The authors claim that "With the proposed evaluation metrics, one may reduce unnecessary machine unlearning operations when data points are determined to be infeasible to unlearn.”, but this is not convincing since erasure is mandatory in any case, and it does not seem reasonable to decide to retrain a model from scratch because a heuristic score ranking method indicates that a single sample may be hard to unlearn. More discussion on the regulatory utility or limitations of unlearnability scores would strengthen this point.
* **Rigorous Unlearning Objective.** The unlearning objective presented in Section 2.1 is based on heuristics, such as maximizing the loss on the forget set, which does not guarantee that an adversary or auditor could not detect the presence of the forget data in the unlearned model. A more rigorous definition of unlearning – one that establishes a statistical similarity to retraining from scratch – would better support the authors’ contributions and align their methodology with recent work in statistically grounded unlearning, e.g., see (Guo et al. 2020).
* **Baseline Comparisons and Additional Techniques.** While the inclusion of KSD is interesting, the paper would benefit from a broader comparison with baselines like influence functions (Koh and Liang, 2017), which are efficient and widely applicable to different architectures. Additionally, incorporating more advanced unlearning techniques or defenses against membership inference attacks (Carlini et al. 2022) would strengthen the empirical evaluation, as only three unlearning algorithms are tested here, limiting the generalizability of results.

### References

Koh and Liang (ICML 2017). Understanding Black-box Predictions via Influence Functions.

Guo et al. (ICML 2020). Certified data removal from machine learning models.

Carlini et al. (S&P 2022). Membership Inference Attacks From First Principles.

**Questions:**

* **Impact of Unlearnability Scores:** Can the authors elaborate on practical applications of unlearnability scores? For example, could these scores help in refining existing unlearning methods to improve handling of difficult samples, or are there contexts in which they could aid in privacy-preserving model design?
* **Empirical Limitations:** What criteria were used to select the three unlearning techniques in the empirical evaluation? Could the authors comment on the generalizability of their methodology to other unlearning frameworks and provide insights on adapting it to handle more complex attack models?

---

> ### Author Response · Authors · 2024-11-22
>
> Thank you very much for your feedback. Below is our response to the concerns and questions you raised.
>
> #### **Regulatory Implications**
>
> This paper aims to introduce a new research direction focused on investigating the feasibility of unlearning. to understand the factors that influence the feasibility of unlearning data, specifically in a manner that is agnostic to any particular unlearning algorithm. The main focus is on the evaluating the feasibility of unlearning and understanding the relation through the lens of KSD based scoring, but we didn't invest in designing unlearning algorithm using the KSD scoring even the potential exist.
> The regulatory implications of "machine unlearning" in relation to the "feasibility of unlearning" are beyond the scope of this research and are left for future exploration. We encourage further research to investigate these aspects in greater detail.
>
>
> #### **Rigorous Unlearning Objective**
>
>
> To ensure that unlearned data is removed from the model, we explored the Membership Inference Attack efficiency. MIA efficacy is quantified by the ratio of samples predicted as "forgotten samples" (True Negatives \textit{TN}) to the total number of samples in the forgetting set $|\mathcal{D}_f|$. The MIA-efficacy reflect the effectiveness of unlearning, where higher MIA-efficiency implies less information about samples unlearned $D_f$ , indicating more successful unlearning outcome. This criteria is reported in Table 2., and full statistical analysis in Table 8.
>
> #### **Baseline Comparisons and Additional Techniques**
>
> The primary contribution of this paper lies in exploring the "Feasibility of Machine Unlearning" before diving into investigating the unlearning algorithm. Previous works have rushed into providing the new machine unlearning approaches, without solid understanding about the unlearning feasibility of data. However, this research direction has a great potential to invite future research into the feasibility of unlearning, providing a robust basis for algorithm development.
>
> Although the Influence function is one of the most mathematically solid base feature attribution method, it is significantly expensive and incomparable w.r.t the KSD which has great potential for understanding and exploring the data-model distribution.
>
> #### **Impact of Unlearnability Scores**
>
> As discussed later, the primary goal of this study is to understand the factors contributing to the feasibility of data unlearning, with particular emphasis on investigating this feasibility challenge independently of specific unlearning algorithms. The practical application of KSD-based scores for implementing unlearning algorithms lies beyond the scope of this research and is left for future exploration. Instead, the focus is directed towards evaluating the feasibility of unlearning and analyzing its relationship through the perspective of KSD-based scoring. While the potential for designing unlearning algorithms using KSD-based scoring exists, our research does not pursue that direction.
>
> Additionally KSD is measured as the $\mathbb{E}_{x, x' \sim q} [\kappa_p(x, x')]$ any changes to the data samples (unlearning and removing any subset of $\kappa_p$ from the data from) damage the calculated KSD. We employ the KSD as the measurement of model distribution, therefore, we can estimate each of their contribution. Removal of any subset of data will significanly damage the KSD meaning. It is not trivial to  apply the KSD for unlearning; hwoever, the insight for the potential approach on employing KSD for unlearning is only using the Scoring Heuristic to select the easy and difficult samples for unlearning.
>
> #### **Empirical evaluation**
>
> From the literature[1, 2], we noticed that these two methods are most common practice and considered to be the most effective unlearning algorithms. Rather than tailoring our approach to a specific unlearning algorithm, we aimed to highlight an overlooked challenge in previous studies and establish a new research direction.
> Still as part of our evaluation, we conducted Membership Inference Attack to investigate whether the unlearned model carries the influence of unlearned sample after the unlearning and how it varies between an easy vs difficult to unlearn sample.
>
> We hope that in the future research, the potential of KSD based scoring metric be employed for the evaluation of privacy and adversarial attacks.
>
>
> [1] Model Sparsity Can Simplify Machine Unlearning. Jinghan Jia et al. NeurIPS 2023.
>
> [2] Gundavarapu, Saaketh Koundinya, et al. "Machine Unlearning in Large Language Models." _arXiv preprint arXiv:2405.15152_ (2024).

---

> > ### Comment · Reviewer_V6u3 · 2024-11-25
> >
> > I thank the authors for their rebuttal and would like to maintain my original score.

---

### Official Review · Reviewer_4fRu · 2024-11-03

**Soundness:** 2
**Presentation:** 3
**Contribution:** 2
**Rating:** 5
**Confidence:** 3

**Summary:**

The paper introduces a set of KSD-based metrics for quantifying forget difficulty, taking into account the characteristics of the target model and data distribution. It introduces a KSD-based heuristic approach to assess forget difficulty, where KSD is a parameterized kernel function tailored for each model and dataset. These metrics hold significant practical value in supporting decision-making processes.

**Strengths:**

1. The KSD-based metrics presented in the paper are particularly intriguing, as they offer valuable insights into the relationship between data and models in the machine unlearning field.
2. The paper is easy to follow. The authors have effectively communicated their ideas, making complex topics accessible and engaging for the audience.
3. Understanding which samples are more difficult to unlearn has the potential to aid the development of machine unlearning.

**Weaknesses:**

1. Table 1 shows many counterintuitive numerical results, such as the basic baseline GradAsct achieving 0% accuracy on the forget set while maintaining 99% accuracy on the test set. Even when the authors' metric indicates that the most difficult-to-unlearn samples to forget in SVNH can also achieve 0% accuracy on the forget set, the accuracy on the test set is mostly around 80%. This result is incredibly hard to believe, especially since the current state-of-the-art GradAsct (enhanced GradAsct baseline: NegGrad+ proposed by [A]) cannot achieve such results.

2. The authors claim that the metric proposed in the paper does not rely on a specific unlearning algorithm, making it unreasonable to only select the simplest baseline finetune and GradAsct for the experiments. This suggests that the metric may only be effective for finetune and GradAsct. Considering the existence of different methods such as teacher-student methodology [A], weight saliency [B], knowledge distillation [C], Fisher [D], and Newton Update [E], simple finetune and GradAsct cannot adequately represent these methods. As a primary contribution of proposing some metrics, the authors should select a representative method from various heuristic unlearning works to verify that the metric does not depend on any specific unlearning algorithm. Only when the phenomenon observed in the authors' metric consistently exists across these different methods can it be concluded that the metric does not rely on a specific unlearning algorithm.

3. The author's citations can be quite misleading in several instances. Such as, in lines 98-99, the authors mention: "Gradient Ascent methods (Thudi et al., 2022; Graves et al., 2021.), adjust the model’s weights in the direction of the gradient to increase the model’s error on the data intended for forgetting." However, it's difficult to classify the methods of Thudi et al. (2022) and Graves et al. as ascent, since ascent implies the need to compute the negative gradient, as in [J], rather than merely adjusting the model.
    In line 112, they state, "Guo et al. (2020) [E] introduced the concept of certified unlearning, grounded in information theory and specifically tailored to the Fisher Information Matrix." However, to my knowledge, Guo et al. (2020) do not mention anything related to the Fisher Information Matrix. If the authors intended to reference the Fisher unlearning method, I suspect they meant to cite [D]. Alternatively, if they intended to reference the use of information theory and the Fisher metric to evaluate unlearning methods, I would guess they meant [I].

4. The authors lack descriptions of some baseline settings and the choice of evaluation metrics. Please refer to my question for specifics.



[A] Towards Unbounded Machine Unlearning. Meghdad Kurmanji, et al. NeurIPS 2023.

[B] SalUn: Empowering Machine Unlearning via Gradient-Based Weight Saliency in Both Image Classification and Generation. Fan, Chongyu, et al. ICLR, 2024.

[C] Can bad teaching induce forgetting? unlearning in deep networks using an incompetent teacher. Vikram S Chundawat et al. AAAI 2023.

[D] Eternal sunshine of the spotless net: Selective forgetting in deep networks. Golatkar et al. CVPR, 2020.

[E] Certified data removal from machine learning models. Chuan Guo, et al. ICML, 2020.

[H] Model Sparsity Can Simplify Machine Unlearning. Jinghan Jia et al. NeurIPS 2023.

[I] Evaluating Machine Unlearning via Epistemic Uncertainty. Alexander Becker et al. ECML 2021.

[J] Machine Unlearning of Pre-trained Large Language Models. Jin Yao et al. ACL 2024.

**Questions:**

1. The authors are suggested to explain the mentioned numerical results.

2. Can the proposed metric be applied to [A]-[E]? It can be more convincing if the authors show these in experiments.

3. What is the expression for 'MIA-efficacy'? It would be best to explain what 'MIA-efficacy' is, either in the main text or in the appendix, rather than just directing the reader to a specific paper, as this is not a common MIA evaluation metric (e.g., AUC, attack success rate).

4. Which references did the authors use for the evaluation of GradAsc**, **FineTune, and Fisher?  To avoid confusion, the authors should clarify in line 112 whether these methods are taken from other papers or are their own designs.

5. What is the overfit_threshold in line 670? The authors should ideally provide a brief description of these baselines and their settings, either in the main text or in the appendix.

6. Have the authors tried any NLP-related tasks? I'm particularly curious about the difficulty of forgetting data in NLP compared to CV tasks.

---

> ### Author Response · Authors · 2024-11-22
>
> Thank you very much for your feedback. Below is our response to the concerns and questions you raised.
>
> #### **Experimental results**
> To assess the effectiveness of the proposed scoring metrics and to explore whether the feasibility of unlearning differs across data points, we selected the top five easiest and the top five most difficult samples to unlearn, as determined by the rankings from each scoring metric. The primary focus of our experimental evaluation is on unlearning individual data points, with the corresponding results presented in Table 1.
>
> We have conducted the GradAsct in very careful and controlled process. The parameter setting are given in the Appendix - Table 4. Also the scoring metrics ranks datapoints based on the closeness to the decision boundary and similar samples which influence on the difficulty of unlearning.
>
> All of unlearning algorithms have been tested on the top 5 easy / difficult samples for 5 times with 5 random seed to avoid any cherry picking. The reported unlearning accuracy in Table.1 are only for individual sample unlearning.
>
> 1- Unlearning Accuracy:
> -   Easy
> 	-   Majority of unlearning algorithms achieved zero accuracy on the forget set $S_F$ (Figure 2, Appendix Table 6.)
> 	-   The Predictive accuracy of model on the remaining $S_R$ and test $S_T$ subsets almost remained similar to the original model [Table 6]
> 	  - Exact unlearning (retraining) of an easy sample is achievable (Table 1, Forget Accuracy)
>   -   Difficult
> 	  - The models' performance was jeopardized (reduction on accuracy by nearly 50 percent on test data for CIFAR10 // Figure. 2., Table. 1. ) or didn't result in successful unlearning (no change to the accuracy of model on the forget set Table 1) (zero difference before and after unlearning )
>
>
> #### **Additional unlearning baselines**
>
> We have conducted the experiments on the Fisher algorithm and the results are reported in the appendix. We also conducted the exact unlearning (retraining from scratch ) to compare the scoring metrics and ensure that our assessments of easy / difficult to unlearn is align with the exact unlearning. From the literature[H, K], we noticed that these two methods are two of the most common practice and effective unlearning algorithms. We didn’t want to engage with tailoring our approach to unlearning algorithm. Our goal is introducing an existing challenge that has been ignored by the previous methods and creating a new branch of research.
>
> We emphasize that the primary contribution of this paper is the investigation of the "feasibility of machine unlearning" before diving into investigating the unlearning algorithm. Previous works have rushed into providing the new machine unlearning approaches, without solid understanding about the feasibility of data. However, this research direction has a great potential to invite the future research on the feasibility of unlearning.
>
> #### **Expansion of Scoring metrics application**
> This paper focuses on positioning the concept of unlearning within the broader research landscape, and aims to rise the community awareness on the challenges reserved for this problem. We discussed the factors influencing the feasibility of unlearning on data samples and introduce a KSD-based scoring metric that is independent of any specific unlearning algorithm. Also the six factors that are contributing to the unlearning difficulty can also be evaluated for any unlearning algorithms.
> From the feedback highlighted from reviewers, we can observe the potential for a novel line of research on the feasibility of unlearning. The main contribution of this research is the introduction of feasibility of unlearning for the. Our paper is the baseline for the future research on the feasibility of unlearning and the researchers can employs our scoring metrics to the other methods.
>
>
> #### **Membership Inference Attack efficacy**
> We are thankful for your recommendation. We will address your feedback in the revised version. Here is the statement of how we calculated the “MIA efficacy”.
>
> The criteria is quantified by the ratio of samples predicted as "forgotten samples" (True Negatives \textit{TN}) to the total number of samples in the forgetting set $|\mathcal{D}_f|$. Since the post-unlearning, the model $\theta_u$ should have effectively "forgotten" the information related to the samples in the forgetting set.

---

> ### Author Response · Authors · 2024-11-22
>
> #### **Reference Clarification**
>
> For the experimental evaluation and the unlearning algorithms, we adhered strictly to the design framework recommended by [H]. We wanted to ensure that the validity of our experimental evaluations and guarantees that our KSD-based scoring metrics remain entirely independent of the specific unlearning settings.
>
> #### **Overfitting threshold**
>
>
> The overfitting threshold is a mechanism introduced specifically for GradAscent during the unlearning process. Without this control, GradAscent can lead to a significant increase in the error rate, rendering the results invalid. To address this issue, we define a cap on the model's error for the forget set, referred to as the _overfitting threshold_. This threshold prevents the unlearning loss from becoming excessively high, which mitigates the risk of distorting the unlearned model. In particular, unlearning a single data point with GradAscent requires careful control to ensure the process is effective. By applying this threshold cap, we aim to prevent excessive error growth while preserving the overall quality of the model.
>
>
>
> #### **Feasibility of unlearning for NLP**
>
> At this stage of our research, we have completed experimental evaluations on the image classification dataset. As a subsequent step, we aim to extend our investigation to examine the feasibility of unlearning in natural language models and their corresponding datasets.
>
> #### **Citations**
> We are so thankful for your feedback, we addressed them in the revised version of the paper.
>
>
> #### References
>
> [A] Towards Unbounded Machine Unlearning. Meghdad Kurmanji, et al. NeurIPS 2023.
>
> [B] SalUn: Empowering Machine Unlearning via Gradient-Based Weight Saliency in Both Image Classification and Generation. Fan, Chongyu, et al. ICLR, 2024.
>
> [C] Can bad teaching induce forgetting? unlearning in deep networks using an incompetent teacher. Vikram S Chundawat et al. AAAI 2023.
>
> [D] Eternal sunshine of the spotless net: Selective forgetting in deep networks. Golatkar et al. CVPR, 2020.
>
> [E] Certified data removal from machine learning models. Chuan Guo, et al. ICML, 2020.
>
> [H] Model Sparsity Can Simplify Machine Unlearning. Jinghan Jia et al. NeurIPS 2023.
>
> [I] Evaluating Machine Unlearning via Epistemic Uncertainty. Alexander Becker et al. ECML 2021.
>
> [J] Machine Unlearning of Pre-trained Large Language Models. Jin Yao et al. ACL 2024.
>
> [K] Gundavarapu, Saaketh Koundinya, et al. "Machine Unlearning in Large Language Models." _arXiv preprint arXiv:2405.15152_ (2024).
>
> [L] Jamie Hayes, Ilia Shumailov, Eleni Triantafillou, Amr Khalifa, and Nicolas Papernot. Inexact unlearning needs more careful evaluations to avoid a false sense of privacy, 2024.

---

> ### Comment · Reviewer_4fRu · 2024-11-27
> **Thank you for the author's Official Comment**
>
> Thanks for the authors' responses. I still have some concerns.
>
> - As an experimental work proposing a metric, **I remain concerned that focusing only on simple baselines such as Finetune and Gradient Ascent might restrict the applicability of the metric.** The author claims, "We didn’t want to engage with tailoring our approach to unlearning algorithms," but conducting experiments only on these simple baselines is already tailoring your approach to the Finetune and Gradient Ascent algorithms. This could limit the paper's contribution to something along the lines of *"Towards Understanding the Feasibility of Finetune/Gradient Ascent Unlearning."* I believe that incorporating other actively researched methodologies, such as teacher-student frameworks [A], weight saliency [B], and Newton Update [E], would significantly strengthen the experiments. At the very least, like [H], the paper should test Newton Update [E], and the results of these methods should be presented in the main text rather than the appendix.
> - The response did not explain why the GradAsct method in the paper achieves such strong performance, even on the most difficult samples to forget (reaching 0% accuracy on the forgetting set of the SVHN dataset while maintaining an accuracy of up to 80% on the test set). The results are too good to be reconciled with my own experience and outperform the results of NegGrad+ proposed by [A]. **I would like to express concern that the authors have not conducted randomness experiments** to demonstrate that the GradAsct method consistently achieves the claimed results.
> - **I still have concerns about the paper's literature review.** I suggest that the authors carefully refine Section 2.2. I have reviewed the authors' updated paper again and still found some inaccuracies in the description of past literature, which could mislead readers who are not familiar with the machine unlearning field.
>   - For example, in line 106, the goal of [D] is not "distinguishing the forget set from the remaining dataset," but rather ensuring that the scrubbing function $S(w)$ produces a model indistinguishable from one that has never seen the forget set (retraining from scratch), by minimizing the KL divergence between their distributions.
>   - In line 112, it should not say "ensure a high probabilistic similarity between models before and after unlearning," but rather "ensure a high probabilistic similarity between the retraining-from-the-scratch algorithm and the unlearning algorithm." Please note that these two are fundamentally different definitions.
>   - Additionally, Mehta et al. is repeated in line 106, and Izzo is repeated in line 111.
> - Lastly, as an optional suggestion, since research on LLM unlearning has recently gained momentum, incorporating NLP tasks into the experiments, whether in the main text or the appendix, could further validate the proposed metric and enhance the paper's contribution and impact. I also suggest the authors will discuss how these ideas can help the machine unlearning community develop algorithms, rather than empirically stating which samples are difficult to forget. Nevertheless, this suggestion will not affect my scoring.
>
> I believe the motivation behind this work is valuable, but the quality of the paper is not sufficient to meet the ICLR acceptance standards, so I maintain my score.

---

### Official Review · Reviewer_ybdG · 2024-11-04

**Soundness:** 3
**Presentation:** 2
**Contribution:** 2
**Rating:** 5
**Confidence:** 5

**Summary:**

This paper considers the problem of determinining the feasibility of machine unlearning. This is done by (a) determining which are the easiest and hardest samples to unlearn based on metrics related to kernel Stein discrepancy (b) unlearning these samples using different unlearning algorithms and (c) see how this impacts the accuracy on the data to be forgotten as well as the test set.

Overall the experiments are solid, but what I found lacking from the paper is discussion and understanding of the significance of the results.

**Strengths:**

Machine unlearning, even though well studied is not well-understood -- mostly because it is usually not well-defined. The problem studied is thus definitely well-motivated.

I feel like what the paper is trying to get at here is "in-distribution" and "out-of-distribution" samples -- in-distribution being those samples that are very close to, or combinations of the rest of the data, while out-of-distribution samples being outliers or others.  In general, one would expect the latter to be easier to unlearn. In addition, it is also unclear why unlearning in-distribution points should lead to lower performance on them -- for example, if we can classify a typical zero accurately from a classifier trained without this zero. A lot of prior work has ignored these subtleties in the definition and practice of unlearning, and this work does attempt to throw light on them.

**Weaknesses:**

1. The major weakness of the paper is that it does not offer much by way of discussion and conclusion from the experiments. The experiments are presented in form of tables, with a short discussion section about different algorithms and metrics, but at the end we do not learn much about what we learn overall from the exercise, and why we get to see what we see. Adding a proper discussion section that tries to explain the results would significantly improve the paper.

2. It is unclear to me why so many different variants of kernel Stein discrepancy are needed as they appear to needlessly complicate the message. Is it because the different measures emphasize different aspects? What kind of aspects?

3. The paper would be improved by adding references to other work that questioned model-based unlearning -- see, for example, [1] https://www.usenix.org/conference/usenixsecurity22/presentation/thudi

**Questions:**

See above.

---

> ### Author Response · Authors · 2024-11-22
>
> Thank you very much for your feedback. Below is our response to the concerns and questions you raised. Also according to your recommendation, the new citation is added in the revised version of paper.
>
>
> #### **Discussion on Unlearning experimental results**
>
> - Accuracy:
> 	- Easy
> 		- Majority of unlearning algorithms achieved zero accuracy on the forget set $S_F$ (Figure 2, Appendix Table 6.)
> 		- The Predictive accuracy of model on the remaining $S_R$ and test $S_T$ subsets almost remained similar to the original model [Table 6]
> 	  - Exact unlearning (retraining) of an easy sample is achievable (Table 1, Forget Accuracy)
>   -   Difficult
> 	  - The models' performance was jeopardized (reduction on accuracy by nearly 50 percent on test data for CIFAR10 // Figure. 2., Table. 1. ) or didn't result in successful unlearning (no change to the accuracy of model on the forget set Table 1) (zero difference before and after unlearning )
>
> - Unlearning Loss
>
> 	- For an easy to unlearn sample
> 		-   The model Error on the forget set has increased Table. 1. & Table 7.
> 	    -   A noteworthy observation from comparing accuracy Table 6, and loss Table 7 is that for the easy-to-unlearn cases, when the unlearning process appears to have minimal impact on the model's predictive capability on the forget set. The models maintain the ability to predict these datapoints even after unlearning. However, including unlearned models' loss analysis of the loss table reveals that, despite the sustained accuracy, the unlearning process increases the models' loss on the forget set for easy-to-unlearn datapoints. This suggests that while the model can still predict the forget datapoints, the error threshold for flipping a datapoint’s label remains high.
>
> 	-   Difficult
> 		-  Unlearning has minimal to no impact on the model's error on the forget set, indicating that these datapoints are more resistant to unlearning. (Table. 7 )
> 		- For cases the unlearning negatively jeopardized the model accuracy, the unlearned model error is also reflected on the loss
>
> - “Distance of Parameter Shift (DPS)”
>
> 	- The layer-wise distance between the original model and the unlearned model is shown in Table 5. Among the scoring metrics, EMSKSD exhibits the smallest distance on the model’s parameters, indicating that it causes less disruption during unlearning. On the other hand, SSN produces the largest gap, as anticipated. This is because SSN tends to select data points near the decision boundary, which typically have larger gradient magnitudes, thereby increasing the distance between the original and unlearned models.
>
>
> - “Resistance to Membership Inference Attack (MIA)”:
>
> 	- The  "easy" samples identified by EMSKSD consistently show higher MIA-efficacy, whereas "difficult" samples often with lower MIE-efficacy which clearly indicate the influence of difficult was not unlearned from the model.

---

> ### Author Response · Authors · 2024-11-22
> **Official Comment by Authors (Pt. 2)**
>
> #### **Unlearning difficulty and Scoring metrics**
>
> The the unlearning difficulty factors are categorized into two major groups namely 1) data points with/without strong ties (factor 1, 4-6) and 2) predictive confidence (factor 2-3). Our aim is to develop a unlearning difficulty scoring metric that jointly considers these two classes of factors.
> Two of those factors are purely dependent to the data-model distribution and are determined before unlearning the model.
>
> - MKSD: evaluates both the immediate proximity of neighboring data points and the degree of strong similarities as reflected by elevated Stein Kernel values. A higher MKSD score indicates greater similarity and a larger "resistance set," meaning that a larger portion of the training data would need to be unlearned alongside the target data point. This scenario is typically undesirable as it increases the complexity of unlearning.
>
> - MSKSD: The sum of the Stein Kernel values for each data point generally provides an indication of strong similarities with other samples within the dataset. However, this measurement can sometimes be misleading if negative values from other samples overshadow the positive similarities. This can result in positive and negative values negating each other. For this issue, we employ a standardization approach to the Stein Kernel values for each data point, denoted as $\kappa_{\theta}((\mathbf{x}_i, y_i), (\cdot, \cdot)$. By standardizing these values, we can prevent the negation effect, and sum the exponential values of the standardized Stein Kernels to properly value the positively correlated samples and avoid their cancellation with negative values.
>
>
> - SSN: We propose that data points with high Stein Score Norms (SSN) are typically located further from the dense centers of their respective classes and closer to the decision boundary. The Stein Score, defined as $\nabla_a \log P_\theta$ is larger for samples near the decision boundary, making such points prime candidates for unlearning. These data points are evaluated and ranked based on the magnitude of their Stein Score vectors, These points are evaluated and ranked based on the magnitude of their Stein Score vectors $\nabla_{\theta} \log P_{\theta}(\mathbf{x}_i, y_i)$ . By identifying and prioritizing data points with the highest Stein Score Norms, we can efficiently target samples that are most susceptible to unlearning due to their proximity to the decision boundary.
>
> - EMKSD In investigating unlearning algorithms, combining the uncertainty of the model's prediction with similarity data points.

---

### Official Review · Reviewer_R2kM · 2024-11-11

**Soundness:** 3
**Presentation:** 2
**Contribution:** 3
**Rating:** 5
**Confidence:** 3

**Summary:**

This paper studies how to estimate the difficulty of effectively unlearning training data points from models. In Section 3, it summarizes six main factors that can impact the effectiveness of machine unlearning, including the size of the unlearning expansion, resistance to membership inference attacks (MIA), distance to the decision boundary, tolerance of performance shift, number of unlearning steps, and the distance of parameter shift. It then groups these factors into two categories: the existence of strong ties among data points and predictive confidence. In Section 4, the paper introduces the notion of kernel strain discrepancy (KSD) and four potential variants to convert KSD into aggregated pairwise kernel values for each data point. In Section 5, it conducts empirical evaluations to observe the following phenomena: i) the relationship between KSD-based scores and the predictive performance of the unlearned model; ii) the effectiveness comparison among the four variants; and iii) the effectiveness comparison of unlearning algorithms against hard-to-unlearn data points.

**Strengths:**

+++ This paper studies the variations in difficulty among different training data points, offering an interesting and novel perspective in machine unlearning.

++ It introduces KSD-based scores to measure unlearning difficulty.

+ Experiments are conducted on two CNN models and three image datasets to empirically investigate the effectiveness of the KSD-based scores.

**Weaknesses:**

--- The paper does not provide sufficient discussion on how to incorporate the KSD-based scores into the overall machine unlearning workflow. It also lacks results on the efficiency of computing these scores. Providing both the computational complexity and empirical evaluations of computational efficiency would strengthen the work. Additional discussion would also be valuable. For instance, could a new unlearning-difficulty-aware algorithm be developed to leverage the KSD-based scores for more effective and efficient machine unlearning? Alternatively, if computing the KSD-based scores is comparable in cost to running certain unlearning algorithms, it would be helpful to clarify how these scores could further enhance the machine unlearning process.

-- It is unclear how the KSD-based scores relate to the six difficulty factors. Ideally, the experiments should first verify that these six factors consistently represent the unlearning difficulty of different samples. Currently, however, the experiments lack systematic results on the relationship between unlearning difficulty and the six factors, which makes the effectiveness of these factors unconvincing. Establishing a solid relationship between the six factors and unlearning difficulty—demonstrating that the factors truly correspond to unlearning difficulty—would allow the paper to propose a unified, holistic metric of unlearning difficulty based on these factors. With this unlearning difficulty metric in place, the paper could then systematically validate that the KSD-based scores indeed reflect unlearning difficulty. The current factor-by-factor approach to evaluating unlearning difficulty (with some factors omitted or combined) is insufficient to convincingly verify that the KSD scores truly capture unlearning difficulty.

**Questions:**

1. Is it possible to provide efficiency results of the KSD-based scores?

2. Is it possible to leverage the KSD-based scores to develop unlearning-difficulty-aware algorithms for more effective machine unlearning?

3. Is it possible to develop a more unified and holistic metric for the unlearning difficulty?

4. Is it possible to provide more systematic empirical results to verify that the KSD-based scores indeed reflect the unlearning difficulty in terms of all six factors?

---

> ### Author Response · Authors · 2024-11-22
>
> Thank you very much for your feedback. Below is our response to the concerns and questions you raised.
>
> #### **Computational complexity**
>
> The computation complexity of MKSD, MSKSD, and EMSKSD is bounded by $O(n \times g^2)$ as aggeragation happens over all data which is multiplied by the complexity of $\kappa_\theta$ if bounded by gradient of $\nabla_a P_\theta$
>
> For SSN, the computation complexity is bounded by the $\nabla_a P_\theta$ => $O(g)$.
>
> For the experimental evaluation, we will conduct the experiments on the samples data and we will report the following experimental results in the revised version.
>
>
> #### **Difficulty aware unlearning algorithm**
>
> In this paper, the main goal is introducing a new research direction for investigating the feasibility of unlearning. Our intention is understanding the contributing factor on the unlearning feasibility of data and more importantly investigating this challenge agnostic to unlearning algorithms. Development of "Difficulty Aware Unlearning algorithm" seems a valubale and most immidiate application  of our research, it is out of scope of our work and can be left for the future research. The main focus is on the evaluating the feasibility of unlearning and understanding the relation through the lens of KSD based scoring. We didn't invest in designing unlearning algorithm using the KSD scoring even the potential exist.
>
> Additionally KSD is measured as the $\mathbb{E}_{x, x' \sim q} [\kappa_p(x, x')]$ any changes to the data samples (unlearning and removing any subset of $\kappa_p$ from the data damage the calculated KSD.
>  We employ the KSD as the measurement of model distribution, which helps us to estimate each of their contribution. Removal of any subset of data will significantly damage the true meaning of KSD. It is not trivial to apply the KSD for unlearning; however, the insight for the potential approach on employing KSD for unlearning can be noted as "employing the Scoring Heuristic to select the easy and difficult samples for unlearning".
>
>
> ####  **Unlearning Feasibility and Scoring Metrics**
>
> We categorized the difficulty factors into two major groups namely 1) data points with/without strong ties (factor 1, 4-6) and 2) predictive confidence (factor 2-3). Our aim is to develop a unlearning difficulty scoring metric that jointly considers these two classes of factors.
>
>
> For example, Size of Unlearning Expansion, as stated in [1] (Section.1 Right Column, 3rd Paragraph) in real world scenarios to forget a requested sample we need to unlearn the whole class. The correlation between similar samples, combined with the generalization strength of deep neural networks, can act as a significant resistance to unlearning a specific sample. The correlation and similarity among samples are inherently dependent on the model's distribution. The **Stein Kernel** provides a mechanism to quantify these correlations, conditioned on the model distribution. As illustrated in [1] (Section A.2, Figure 3), the pairwise Stein Kernel values between a target data point and its similar samples reveal how related samples are distributed across various classes. Typically, data points exhibit strong Stein Kernel correlations with relevant samples within the same class, while their similarity to samples from other classes diminishes significantly. This ability to distinguish intra-class and inter-class similarities underscores the utility of the Stein Kernel in understanding unlearning dynamics.
>
> Geometric Distance to Decision Boundary: As stated in [1,2] (Section 1, Right Column, 3rd Paragraph) samples with the highest uncertainty are typically located closest to the classification boundary. Building on the ideas presented in [1], we infer that unlearning samples near the decision boundary is relatively easier compared to those situated at the center of clusters, which are more strongly tied to adjacent data points.
>
> In [3] (Section 1, Page 2, 2nd Paragraph), it is discussed that robust classifiers tend to learn geometrically more complex decision boundaries. These robust models often assign significantly lower confidence scores to low-density samples near the boundary (Section 4.2, Page 10). Similarly, [4] highlights that data points farther from the decision boundary tend to have higher confidence, while those closer to the boundary exhibit lower confidence.
>
> The Kernel Stein Discrepancy (KSD) plays a critical role in identifying this phenomenon. According to Formula 4, the Stein Score—defined as the gradient of the log probability density function—exhibits higher values for samples closer to the decision boundary, making it an effective metric for capturing and analyzing this concept.

---

> ### Author Response · Authors · 2024-11-22
> **Official Comment by Authors (Pt. 2)**
>
> #### **Unlearning Feasibility: Insights from Experiments and KSD Scoring**
>
>
> Systemic analysis of easy - difficult samples recommended by KSD-based scores are tied to the six unlearning difficulty factors. How the evaluation of six factors differ between an easy vs difficult to unlearn:
>
>
> The experimental results:
>
>  Predictive “Performance Shift”:
> - Accuracy:
> 	-   Easy
> 		-   Majority of unlearning algorithms achieved zero accuracy on the forget set $S_F$ (Figure 2, Appendix Table 6.)
> 		-   The Predictive accuracy of model on the remaining $S_R$ and test $S_T$ subsets almost remained similar to the original model [Table 6]
> 		  -   Unlearning by retraining of a single easy datapoint can achieve the zero accuracy on the forget set
>   -   Difficult
> 	  - The models' performance was jeopardized (reduction on accuracy by nearly 50 percent on test data for CIFAR10 // Figure. 2., Table. 1. ) or didn't result in successful unlearning (no change to the accuracy of model on the forget set Table 1) (zero difference before and after unlearning )
>
> - Unlearning Loss
>
> 	-   Easy
> 		-   The model Error on the forget set has increased Table. 1. & Table 7.
> 	    -   A noteworthy observation from comparing accuracy Table 6, and loss Table 7 is that for the easy-to-unlearn cases, when the unlearning process appears to have minimal impact on the model's predictive capability on the forget set. The models maintain the ability to predict these datapoints even after unlearning. However, including unlearned models' loss analysis of the loss table reveals that, despite the sustained accuracy, the unlearning process increases the models' loss on the forget set for easy-to-unlearn datapoints. This suggests that while the model can still predict the forget datapoints, the error threshold for flipping a datapoint’s label remains high.
>
> 	-   Difficult
> 		-   Unlearning has minimal to no impact on the model's error on the forget set, indicating that these datapoints are more resistant to unlearning. (Table. 7 )
>
> - “Distance of Parameter Shift (DPS)”
>
> 	- The layer-wise distance between the original model and the unlearned model is presented in Table 5. Among the evaluated scoring metrics, **EMSKSD** demonstrates the smallest distance in the model’s parameters, indicating that it induces minimal disruption during the unlearning process. In contrast, **SSN** produces the largest distance, as expected. This result is attributed to SSN’s tendency to prioritize data points near the decision boundary, which generally have larger gradient magnitudes. Consequently, this leads to a greater divergence between the original and unlearned models.
>
>
> - “Resistance to Membership Inference Attack (MIA)”:
>
> 	- The  "easy" samples identified by EMSKSD consistently show higher MIA-efficacy, whereas "difficult" samples often with lower MIE-efficacy which clearly indicate the influence of difficult was not unlearned from the model.
>
>
> #### **Unification of unlearning factors**
>
> From the unlearning difficulty factors we categorize them  into two major groups 1) data points with/without strong ties (factor 1, 4-6) and 2) predictive confidence (factor 2-3).Both of these characteristics are embedded from the original KSD formula (Formula 4.). The KSD formula unifies the raw feature similarity, closeness to the decision boundary (in terms of score similairy)  and Mutual Influence of Prediction Shifts. Later the KSD-based scoring metric enhances these factors to ensure that all major controbuting factos for unlearning are considered effectively.
>
>
>  [1] Chen, Min, et al. "Boundary unlearning: Rapid forgetting of deep networks via shifting the decision boundary." Proceedings of the IEEE/CVF Conference on Computer Vision and Pattern Recognition. 2023.
>
> [2] Nguyen, Hieu T., and Arnold Smeulders. "Active learning using pre-clustering." Proceedings of the twenty-first international conference on Machine learning. 2004.
>
> [3] Kienitz, Daniel, Ekaterina Komendantskaya, and Michael A Lones. "Comparing Complexities of Decision Boundaries for Robust Training: A Universal Approach." Proceedings of the Asian Conference on Computer Vision. 2022.
>
> [4] Li, Chen, Xiaoling Hu, and Chao Chen. "Confidence estimation using unlabeled data." arXiv preprint arXiv:2307.10440 (2023).

---

### Author Response · Authors · 2024-11-23

We are immensely grateful for the time and effort invested by all the reviewers. It brings us great pleasure to see that all reviewers unanimously praised the importance of the research question addressed in this paper, which explores a largely overlooked aspect of machine unlearning. Reviewer **R2kM** emphasized the significance of employing Kernelized Stein Discrepancy (KSD) to assess unlearning difficulty.  **V6u3** praised our research direction as an original and timely contribution to the field of unlearning. Reviewer **4fRu** described the use of KSD-based metrics as "intriguing," highlighting their ability to provide valuable insights into the interplay between data and models within the context of machine unlearning.  V6u3 characterized the KSD-based scoring approach as both "innovative" and "technically sound."
Furthermore, **4fRu** noted the paper is easy to follow and the ideas are communicated effectively.

As a general response to all reviewers, we would like to provide a recap on the position of this paper.


**Contribution restatement**

1.  The first attempt to understand the feasibility/difficulty of machine unlearning
2.  The KSD driven heuristic group as the preliminary attempt that works generally well
3.  The new research direction that may interest certain research community

We are deeply thankful for the time and effort each reviewer has dedicated to evaluating our work. We provide individual response to their feedback.

---

### Meta-Review · Area_Chair_qaYX · 2024-12-25

**Metareview:**

The focus of this paper is on understanding the feasibility of machine unlearning with regards to unlearning individual training data. The authors introduce a new metric that can be tailored to the data distribution and model, called the Kernelized Stein Discrepancy. This measure is obtained by using Stein's identity and choosing an appropriate kernel function to compute a notion of discrepancy between two distributions. This measure "quantifies" the difficulty of unlearning individual data and the paper uses existing methods for unlearning to validate this approach. While the reviewers felt that the new concept was interesting, there was inadequate justification and not sufficient conclusions could be drawn from the experimental evaluation. There are also a number of different metrics proposed which makes the message a bit unclear. The authors should take the reviewer comments and act on them in preparing a future version.

**Additional Comments On Reviewer Discussion:**

There was significant discussion during the rebuttal period.

---

### Decision · Program_Chairs · 2025-01-22

Reject